

# Improving maps of forest aboveground biomass: A combined approach using machine learning with a spatial statistical model

Shaoqing Dai [1,2,*], Xiaoman Zheng [1,2,*], Lei Gao [3], Chengdong Xu [4], Shudi Zuo [1,2,5], Qi Chen [6], Xiaohua Wei [7], Yin Ren [1,5]

[1] Key Laboratory of Urban Environment and Health, Key Laboratory of Urban Metabolism of Xiamen, Institute of Urban Environment, Chinese Academy of Sciences, CN 361021, China

[2] University of Chinese Academy of Sciences, CN 100049, China

[3] CSIRO, Waite Campus, Urrbrae, SA 5064, Australia

[4] State Key Laboratory of Resources and Environmental Information System, Institute of Geographic Sciences and Natural Resources Research, Chinese Academy of Sciences, CN 100049, China

[5] Ningbo Urban Environment Observation and Research Station-NUEORS, Chinese Academy of Sciences, CN 315800, China

[6] Department of Geography, University of Hawai'i at Mānoa, Honolulu, HI 96822, USA

[7] Department of Earth and Environmental Sciences, University of British Columbia, Kelowna, BC V1V 1V7, Canada

*These authors contributed equally to this work.

*Correspondence to*: Yin Ren (yren@iue.ac.cn)



**Abstract:** Aboveground biomass (AGB) estimates at the plot level plays a major part in connecting accurate single-tree AGB measurements to relatively difficult regional-scale AGB estimates. However, complex and spatially heterogeneous landscapes, where multiple environmental covariates (such as longitude, latitude, and forest structure) affect the spatial distribution of AGB, make upscaling of plot-level models more challenging. To address this challenge, this study proposes an approach that combines machine learning with spatial statistics to construct a more accurate plot-level AGB model. The study was conducted in a *Eucalyptus* plantation in Nanjing, China. We developed, evaluated, and compared the accuracy and performance of three different machine learning models [support vector machine (SVM), random forest (RF), and the radial basis function artificial neural network (RBF-ANN)], one spatial statistics model (P-BSHADE), and three combinations thereof (SVM & P-BSHADE, RF & P-BSHADE, RBF-ANN & P-BSHADE) for forest AGB estimates based on AGB data from 30 sample plots and their corresponding environmental covariates. The results show that the performance indices RMSE, nRMSE, MAE, and MRE of all combined models are substantially smaller than those of any individual models, with the RF & P-BSHADE combined method giving the smallest value. These results demonstrate clearly that combined models, especially the RF & P-BSHADE model, can improve the accuracy of plot-level AGB models and reduce uncertainty on plot-level AGB estimates or even on large-forested-landscape AGB estimates. These research results are important because they reduce the uncertainty in estimates of the regional carbon balance.

**Keywords:** Aboveground biomass, plot-level model, Machine learning, Spatial statistical model



## 1 Introduction

Accurate maps of aboveground biomass (AGB) provide a solid foundation for sound decision-making in sustainable forest management scenarios, such as reducing deforestation, forest degradation, and greenhouse-gas emissions (Bustamante et al., 2016; Houghton et al., 2009; Mendoza-Ponce and Galicia, 2010). Most AGB maps are constructed based on plot-level estimation models, which are challenging to scale up and can ultimately propagate uncertainty to regional AGB maps. The uncertainty of such regional maps can be attributed to two primary sources: (1) the use of inadequate sampling data to construct the plot level prediction models, and (2) model-dependent uncertainty, including unreasonable model-parameter assumptions and improper model structure (Chen et al., 2015; Gao et al., 2016; McRoberts et al., 2016). The present study mainly focuses on reducing the second source of uncertainty.

An estimated 18%–103% of the uncertainty in AGB mapping can be attributed to model-dependent uncertainty (Djomo and Chimi, 2017; Malhi et al., 2004). Although the allometric model, which is the most popular plot-level model, has produced useful results for forest AGB estimates (Conti et al., 2019; Huang et al., 2019), selection error in plot-level allometric modeling still leads to over 40% uncertainty (Djomo et al., 2016; Fayolle et al., 2013; Chave et al., 2014), and simple or complex forms of the allometric model account for 20%–60% of the uncertainty (Picard et al., 2015).

Many different plot-level prediction models other than allometric models have been applied to constructing accurate AGB maps, including linear models (Andersen et al., 2014; Morel et al., 2012), machine learning models (Chen, 2015; Gleason and Im, 2012), and spatial statistical models (Benitez et al., 2016; Propastin, 2012;Van der Laan et al., 2014). With the development of computer-science techniques and advances in nonlinear biomass modeling, machine learning methods have become prevalent. Traditional parametric methods, which summarize data with a fixed number of parameters based on sample size (e.g., logistic regression and perceptron) (Gao and Hailu, 2012), have difficulty characterizing nonlinear relationships between AGB and multiple environmental covariates. By comparison, nonparametric machine learning algorithms, in which the number of parameters depends on the number of training examples (e.g., K-nearest neighbor, support vector machine, and random forest), are advantageous because they are more elastic and do not restrict variable types, the distribution of predictor variables, or the relationship between response and predictor variables (Lu et



al., 2007). In addition, nonparametric machine learning algorithms may offer higher prediction accuracy
(Frey et al., 2019; Gleason and Im, 2012).
Another group of models frequently used to estimate the relationship between forest AGB and multiple
environmental covariates is based on spatial statistical approaches, including geographically weighted
regression and Kriging (Du et al., 2010; Van der Laan et al., 2014; Viana et al., 2012). Spatial statistical
methods are based on analyses of attribute information, such as spatial location (Schabenberger and
Gotway, 2005). Compared with traditional statistical methods, spatial methods integrate spatial factors
that affect model responses, thus removing the constraints of traditional statistical methods that assume
sample independence (Rangel and Bini, 2010) and improving our understanding of spatial
autocorrelation and heterogeneity (He et al., 2011; Rosenberg and Anderson, 2011).
Although many studies have integrated ground-based plot data, multi-source remote-sensing data (e.g.,
LiDAR and Landsat), and machine learning or spatial statistical methods, the prediction accuracy of
current AGB spatial mapping still suffers from uncertainty (McRoberts et al., 2018; Paul et al., 2016;
Saatchi et al., 2011; Zheng et al., 2004; Jachowski et al., 2013; Zhang et al., 2014). First, existing
studies that used machine learning methods have not considered the spatial heterogeneity of multiple
environmental covariates (such as longitude, latitude, and forest structure), which affects the spatial
distribution of AGB (Babcock et al., 2015; Fassnacht et al., 2014). Second, the assumptions of the spatial
statistical method (e.g., spatial autocorrelation and spatial stratified heterogeneity) may not always apply
to forest AGB.
AGB estimates at the plot level serve as a bridge to connect single-tree AGB measurements to AGB
estimates on a regional scale. Accurate AGB mapping at the plot scale provides a basis for future
upscaling to the regional scale. However, the uncertainty and error propagation inherent in different
prediction models make this process challenging. Allometric models are most commonly used to
construct plot-level AGB models, but they cannot fully capture the complex and spatially
heterogeneous landscapes where multiple environmental covariates (such as longitude, latitude, and
forest structure) affect the spatial distribution of AGB. The objective of the present study is to develop
and evaluate a combined machine learning and spatial statistical method that uses ground-based samples
to improve the prediction accuracy of AGB spatial mapping at the plot level. The proposed method





integrates the nonlinear mapping capabilities of machine learning algorithms [i.e., radial basis function
artificial neural network (RBF-ANN), support vector machine (SVM), and random forest (RF)] with the
spatial autocorrelation and stratified heterogeneous advantages of a spatial statistical model (i.e., the
point estimation model of biased sentinel hospital-based area disease estimation, P-BSHADE) (Xu et al.,
2013). Our aim is to answer two specific questions: (1) What are the differences in prediction accuracy
of AGB maps based on different methods? (2) Can the integration of spatial statistical and machine
learning methods improve the accuracy of AGB models at the plot level? We explore these two
questions by studying an empirical case for predicting an AGB map at a *Eucalyptus* plantation in
Nanjing County, China.
**2 Materials and Methods**
**2.1 Site description**
Nanjing County (117°00'–117°36'E, 24°26'–25°00'N, Fig. 1b) is located in the upstream region of the
Jiulong River in Fujian Province, China. Seventy-four percent (145 009 ha) of the county comprises
forests and 79 346 ha are plantations. The region is affected by the South Asian tropical monsoon climate.
In 2014, the average annual temperature in Nanjing County was 21.1°C, with an annual precipitation of
1700 mm and 340 frost-free days. The major soil type is red soil.
The study area has a complex topography with significantly varying elevation (0–1566 m). Forest
composition, structure, and biomass are spatiotemporally heterogeneous. The main tree species are
*Eucalyptus grandis x urophylla*, *Pinus massoniana*, and *Cunninghamia lanceolata*. Recently, the area of
*Eucalyptus* plantations has increased rapidly, reaching 13 338 ha, which is an increase of 10 862 ha in
one decade.

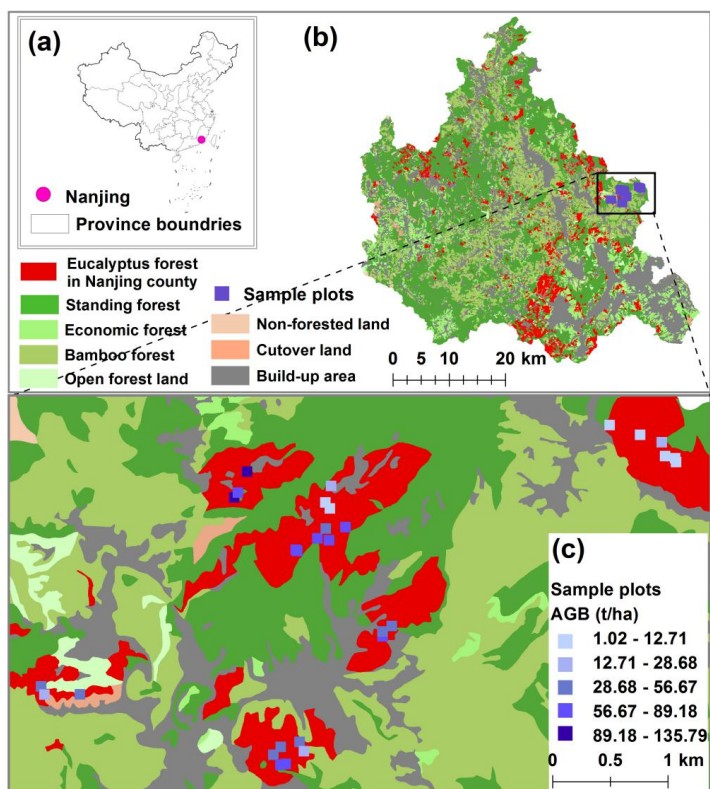

Figure 1. The study area is a typical example of a non-representative–sample problem. (a) Geographical location of the study area. (b) Spatial distribution of *Eucalyptus* plantations (red) and other major forests. (c) Spatial distribution of the 30 sample plots used in this study (blue).

**2.2 Data collection**

**2.2.1 Non-destructive sampling in sample plots**

A total of 30 fixed sample plots were selected in 2012 from the Yongfeng forest farm. The plots were located in the eastern section of the study area (Fig. 1). The 30 sampling plots included ten *Eucalyptus* plantation age groups. In each plot (0.04 ha, 20 m × 20 m), we measured the diameter at breast height (DBH) of all living stems ≥8 cm and the tree height (H). In addition, we measured mean plot-level variables, including stand age, density, longitude, latitude, and altitude.





**2.2.2 Destructive sampling in sample plots: Tree harvest**
Trees were harvested from standard woods in the 30 fixed sample plots. Three trees with a DBH close
to mean DBH of trees in each plot were cut down, for a total of 90 trees harvested from the 30 plots.
We then measured the H and DBH of each harvested tree, as well as the biomass of each organ (foliage,
stems, and branches) to obtain the AGB of each harvested tree. Table B.2 in section S2 of the
Supplementary Material presents the data for the 90 harvest trees. Details on selection of the standard
wood and the cutting process are provided in section S1 of the Supplementary Material.
**2.3 Construction of tree-level allometric models**
All analyses were based on the underlying assumption that the relationship between the response and
predictor variables in the sample data used to construct the models was the same as the relationship in
the entire population. We divided the 90 harvested trees into three age groups (1–2 yr, 3–5 yr, 6–10 yr)
for the tree-level allometric models. The allometric models were then applied to each tree in each
sample plot according to their age, DBH, and H, thereby producing a true measure of AGB for each
sample plot.
**2.4 Construction of plot-level models**
Processing based on model screening was applied to alleviate uncertainty caused by model-dependence
and consisted of the four steps shown in Fig. 2.

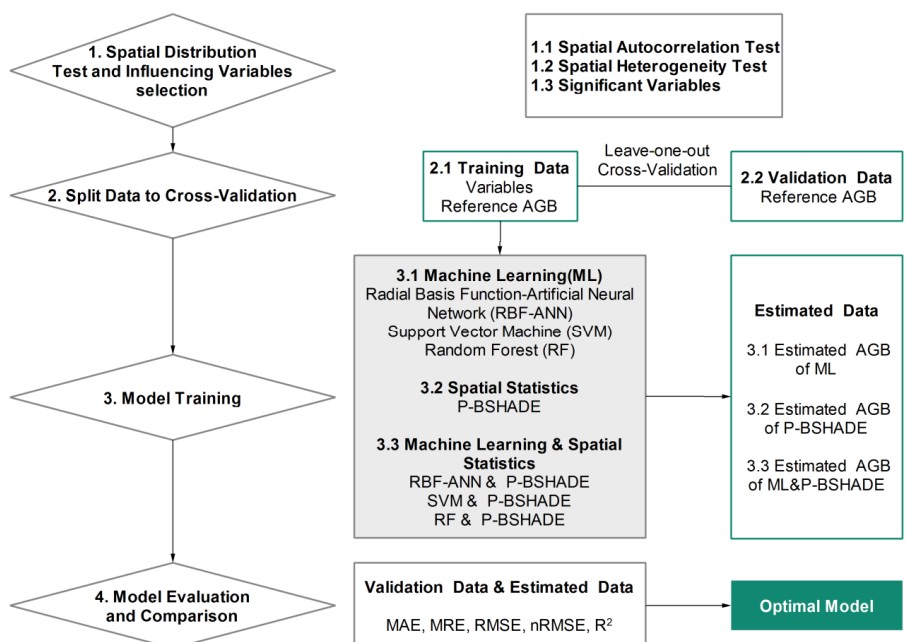


Figure 2. Workflow for screening an optimal model.

**2.4.1 Selection of variables and analysis of resulting spatial distribution**
To create the plot-level model, we first identified predictor variables. Based on our previous work (Ren
et al., 2017), we selected plot-level environmental covariates including longitude and altitude, and forest
attribute variables including forest distribution density, DBH, H, tree stem volume, and forest age.
Pearson's correlation coefficient was used to investigate the correlation between these variables and the
true AGB of sample plots.
We then analyzed the spatial autocorrelation and spatial heterogeneity of AGB data from the selected
sample plots. We used Moran's $I$ (Cliff and Ord, 1981), a commonly used global spatial autocorrelation
index, to evaluate spatial autocorrelation between the true AGBs of sample plots. The spatial stratified
heterogeneity (which refers to the within-strata variance being less than the between-strata variance; it
is ubiquitous in ecological phenomena, such as AGB) of the true AGB of sample plots was evaluated
by using a $q$-statistic generated by applying the GeogDetector model, which is a software tool proposed
by Wang et al. (2016) that analyzes spatial variation of the geographical strata of variables. First, we
used the K-means algorithm to obtain the strata of true AGB for preprocessing by GeogDetector. Next,





we regarded the true AGB as Y, the strata of true AGB as X, and put them into the GeogDetector
model to obtain the $q$-statistic (Wang et al., 2010; Wang et al., 2016).

### 2.4.2 Split datasets

We used the leave-one-out cross-validation method to split the 30 sample plots into 30 sets, with each set
containing two groups of data: (1) validation data (the AGB of one plot) and (2) training data (the AGBs
and predictor variables of the other 29 plots), see Table B.3. The leave-one-out cross-validation method
assumes that, in a dataset containing $n$ samples, each sample serves as a test sample with the other $n-1$
samples serving as training samples. Thus, with $n$ iterations, we can obtain $n$ training datasets and $n$
validation datasets.

### 2.4.3 Model training

Seven models including three machine learning models [Figs. 3(a–3(c)], one spatial statistical model
[Fig. 3(d)], and three combined machine learning and spatial statistical models [Figs. 3(a) and 3(d), 3(b)
and 3(d), and 3(c) and 3(d)] were developed and trained to predict the AGB of sample plots. The three
machine learning models were (a) SVM, (b) RBF-ANN, and (c) RF.
The spatial statistical model (P-BSHADE) required AGB-related variables (reference series). In this
case study, we used the reference-plot AGB data as the variables. The allometric model (Qiu et al.,
2018) was applied to obtain the AGB of each tree in each sample plot. Next, the reference-plot AGB
data consisted of the sum of the AGB of each tree. This method produces the P-BSHADE model shown
in Fig. 3(d). For the combined machine learning and spatial statistical models, the reference plot AGB
data in P-BSHADE were obtained from the results of the SVM [Fig. 3(a)], the RBF-ANN [Fig. 3(b)], or
the RF [Fig. 3(c)]. The three combined models are denoted SVM & P-BSHADE [Figs. 3(a) and 3(d)],
RBF-ANN & P-BSHADE [Figs. 3(b) and 3(d)], and RF & P-BSHADE [Figs. 3(c) and 3(d)]. Each
model was trained on 30 datasets, yielding a total of 30 predicted AGB datasets for 30 sample plots (see
Table B.3, section S2 in the Supplementary Material).



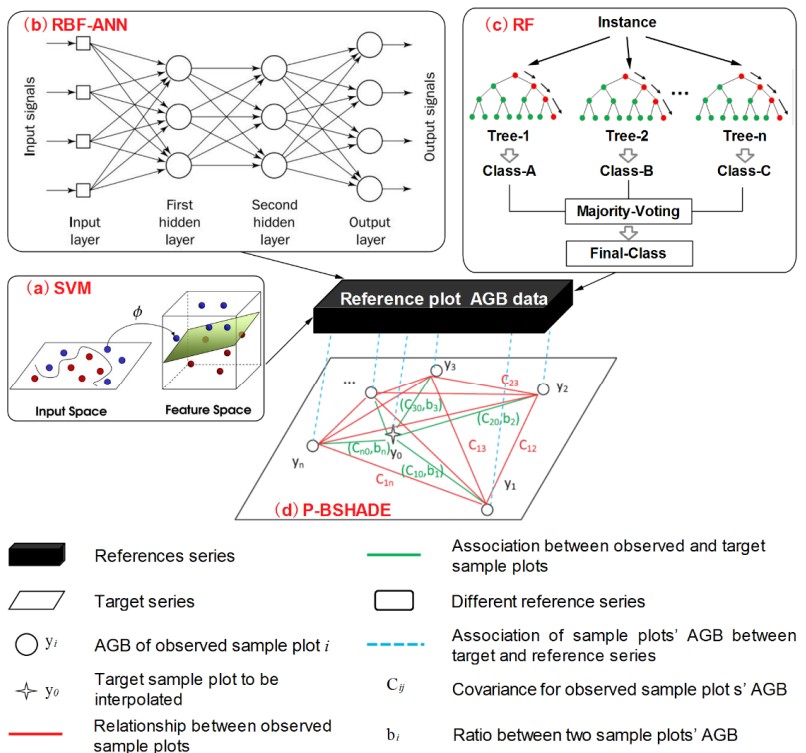



Figure 3. Framework for estimating (a)–(c) the machine learning models, (d) the P-BSHADE model,
and the three models that combine machine learning with the P-BSHADE model (a+d, b+d, c+d).

(1) Machine learning
SVM is a method of supervised learning in machine learning and is often used to solve classification
problems. The basic principle of SVM is to find a hyperplane in the feature space and separate the
positive and negative samples with the minimum misclassification rate (Hearst et al., 1998). RBF-ANN
is a three-layer neural network model, which includes an input layer, a hidden layer, and an output layer.
The transformation from input space to hidden space is nonlinear, whereas the transformation from
hidden space to output space is linear. The function of the hidden layer is to map the vector from the
indivisible low-dimensional linear state to the separable high-dimensional linear state, so as to greatly



accelerate the learning and convergence speed and avoid getting stuck in a local optimum (Elanayar and
Shin, 1994; Xia and Xiu, 2007). RF is a combination of tree predictors such that each tree depends on
the values of a random vector sampled independently and with the same distribution for all trees in the
forest. RF is an effective tool in prediction. Because of the Law of Large Numbers, RF does not overfit.
Injecting the right type of randomness means that RF makes accurate classifiers and regressors (Breiman,

207   2001).

The schematic function for machine learning is
$y_j = f(x_{j,1}, x_{j,2}, x_{j,3}, x_{j,4})$         (1)
where $y_j$ is the AGB of the $j$th sample plot predicted by a machine learning model, $f(...)$ is a machine
learning model represented by a function of $x_{j,k}$ $(k = 1,...,4)$; and $x_{j,1}$, $x_{j,2}$, $x_{j,3}$, and $x_{j,4}$ are the
central longitude, the mean DBH, the mean H, and the forest age of the $j$th sample plot, respectively. A
specific description of the three machine learning models is given in section S1 of the Supplementary
Material.
(2) Spatial statistical model: P-BSHADE
P-BSHADE is an optimal linear unbiased estimation interpolation method based on the assumption of
the simultaneous existence of the spatial autocorrelation and heterogeneity of the target object. We use
it here to solve the problem of an unrepresentative sample imposed by the spatial location of a
convenient sample at the plot level.
The core of the model is to minimize the variances between predicted error and unbiased estimation.
The prediction process of the P-BSHADE model requires strong spatio-temporal coordination between
the predictive variable (forest AGB of target plots) and the reference series (reference forest AGB of
target plots), so as to realize the spatial interpolation of the predictive variable. The model is also a data
fusion approach that combines the observed samples with the reference series (related variable).
P-BSHADE is markedly different from the Kriging and Inverse Distance Weighting (IDW) algorithms.
Compared with Kriging and IDW, the application of P-BSHADE to forest AGB interpolation has
obvious advantages. The spatial distribution of forest AGB is also characterized by spatial
autocorrelation and heterogeneity, which have been taken into account in the P-BSHADE model.



Taking into account spatial heterogeneity can effectively solve the difference in forest AGB
distribution caused by different terrain or geographical location. However, Kriging and IDW only
consider the spatial correlation between plots. In addition, P-BSHADE considers strongly correlated
sample plots as neighboring plots, whereas the Kriging and IDW algorithms consider sites that are
close in proximity.
In brief, the P-BSHADE model includes two steps. First, it obtains reference AGB for all sample plots
by using the allometric model. Second, it uses the reference AGB of the target sample plot and the true
AGB of other sample plots to obtain the weight relationship between the target sample plot and the
other sample plots and puts the true AGB of other sample plots and the weights into Eq. (2) to predict
the AGB of the sample plots. Therefore, positions and distances between plots do not apply here. The
specific mathematical formula for the P-BSHADE model is now described (Hu et al., 2013; Xu et al.,

240    2013).

**a. Objective**
The objective is to interpolate the AGB data of the target sample plot by using data acquired from other
sample plots. A theoretical description is
$\hat{y}_j = \Sigma_{i=1}^{n} w_{ij} y_i$                    (2)
where $\hat{y}_j$ is the AGB of the $j$th sample plot estimated by the P-BSHADE model $(j = 1 - 30, n =$
$30)$; $y_i$ is the true AGB of the $i$th sample plot $(i = 1 - 30, n = 30)$; $w_{ij}$ is the weight (contribution)
of the true AGB of the $i$th sample plot to the AGB to be interpolated of the $j$th sample plot (when $j =$
$1$, $i = 2, 3, ... , 30$; when $j = 1$, $i = 1,\ 3,\ 5, ... , 30)$; $w_{ij}$ is calculated by the true AGB of the $i$-th
sample plot and the allometric model estimation of the AGB in the $j$-th sample plot.
As expected, the estimates of the two properties in Eq. (2) are unbiased:

251                              $E(y_j) = E(\hat{y}_j)$                              (3)

Minimum estimation variance is expressed as

253                              $\min_w \left[ \sigma_{\hat{y}_j}^2 = E(\hat{y}_j - y_i)^2 \right]$                              (4)

where $E$ is the statistical expectation.



**b. Ratio of data from target sample plot to those from other sample plots**
The ratio between data from the target sample plot to those from other sample plots is one of the most
important inputs for estimating the ABG of the target sample plot and is an index of heterogeneity in
the AGB spatial distribution. The relationship between data from the target sample plot and from the
other sample plots is expressed as
$$b_{ij}Ey_j = Ey_i \qquad (5)$$
In most cases, the AGB of any two plots are not equal, and the relationship between them can be
further expressed as the relative bias $b_{ij}$ between the mathematical expectation of $y_j$ and $y_i$.
Considering Eq. (2), Eq. (5) can be written as
$$\sum_{i=1}^{n} w_{ij}b_{ij} = 1 \qquad (6)$$
This equation is generally valid for nonhomogeneous conditions. Clearly, the determination of $b_{ij}$
requires calculating the coefficients $w_{ij}$ $(i = 1, \dots, n, j = 1, \dots, n)$, which is addressed in the following
section.

**c. Weight estimation**
The main challenge in estimation is finding the weights $w_{ij}$ that satisfy the unbiased condition and
that minimize estimation variance:
$$\sigma_{\hat{y}_j}^2 = E(\hat{y}_j - y_i)^2 = C(\hat{y}_j\hat{y}_j) + C(y_iy_i) - 2C(\hat{y}_jy_i) \qquad (7)$$

These weights can be calculated by minimizing the estimation variance and taking unbiasedness into
account:
$$\begin{bmatrix} C(y_1y_1) & \cdots & C(y_1y_n) & b_{1j} \\ \vdots & \ddots & \vdots & \vdots \\ C(y_ny_1) & \cdots & C(y_ny_n) & b_{nj} \\ b_{1j} & \cdots & b_{nj} & 0 \end{bmatrix} \begin{bmatrix} w_{1j} \\ \vdots \\ w_{nj} \\ \mu \end{bmatrix} = \begin{bmatrix} C(y_1y_j) \\ \vdots \\ C(y_ny_j) \\ 1 \end{bmatrix} \qquad (8)$$

where $\mu$ is a Lagrange multiplier. The minimized variance in the estimation error can then be written
as
$$\sigma_y^2 = \sigma_{y_i}^2 + \sum_{i=1}^{n}\sum_{k=1}^{n} C(y_iy_k) - 2\sum_{i=1}^{n} w_{ij}C(y_iy_j) + 2\mu\left(\sum_{i=1}^{n} w_{ij}b_{ij} - 1\right) \qquad (9)$$






The P-BSHADE model is a geospatial model because it has the following characteristics:
1. The P-BSHADE model is mainly based on the assumptions of spatial autocorrelation and spatial
heterogeneity of forest AGB. Therefore, before using P-BSHADE, we first applied the statistical test of
these two theoretical hypotheses (spatial autocorrelation test and spatial differentiation test) for forest
AGB.
2. The prediction process of the P-BSHADE model requires strong spatio-temporal coordination
between the predictive variable (forest AGB of target plots) and the reference sequence (reference
forest AGB of target plots), so as to spatially interpolate the predictive variable.
3. P-BSHADE is an optimal linear unbiased estimation interpolation method that considers temporal
and spatial heterogeneity. Spatial autocorrelation and heterogeneity of AGB data can be added into the
model based on prior knowledge (reference AGB data), following which the linear unbiased optimal
estimation of the target-plot AGB can be obtained by correcting data from a convenient sample plot.
Specifically, for example, the ratio of data from the target sample plot to that from other sample plots is
used [see 2.4.3(2)b section]. In the P-BSHADE model, this ratio plays a very important role in
estimating the forest AGB of the target plots. This ratio is a manifestation of the spatial heterogeneity
of AGB data. P-BSHADE takes into account the reality of the spatial distribution of AGB data and
emphasizes that the spatial distribution of AGB data is heterogeneous.
(3) Combination of machine learning and spatial statistical models
Considering the inherent advantages and disadvantages of P-BSHADE and machine learning, this study
investigates whether their combination can improve the accuracy of forest AGB estimates. Therefore,
P-BSHADE was separately integrated with the three machine learning methods (SVM, RBF-ANN, and
RF) to form three combined models (SVM & P-BSHADE, RBF-ANN & P-BSHADE, and RF &
P-BSHADE). The reference AGBs of the 30 sample plots were replaced by the estimates produced by
the machine learning models. Each combined model was represented as follows:
$\hat{y}_j = \Sigma_{i=1}^{n} w_{ij} y_i$                    (10)





where $\hat{y}_j$ is the estimated AGB of the $j$th sample plot using the combined model ($j =$
$1, 2, \ldots, 30, n = 30$); $y_i$ is the true AGB of the $i$th sample plot ($i = 1, 2, \ldots, 30, n = 30$); $w_{ij}$ is the
contribution in weight of the $i$th true AGB of the sample plot to the $j$th sample plot AGB to be
interpolated (when $j = 1$, $i = 2, 3, \ldots, 30$; when $j = 1$, $i = 1, 3, 5, \ldots, 30$); $w_{ij}$ is calculated by
using the true AGB of the $i$th sample plot and the machine learning estimate of the AGB of the $j$th
sample plot. A detailed description of the combined models and the algorithm formulas is presented in
section S1 of the Supplementary Material.
**2.4.4 Model evaluation and comparison**
To evaluate the accuracy of the AGB estimates of the seven models (SVM, RBF-ANN, RF, P-BSHADE,
SVM & P-BSHADE, RBF-ANN & P-BSHADE, and RF & P-BSHADE), the AGB results were
compared to the reference AGBs of the sample-plot groups (AGB group M in Table B.3). We calculated
four performance indicators, as given by Eqs. (11)–(14) [mean absolute error (MAE), mean relative
error (MRE), root mean square error (RMSE), and normalized root mean square error (nRMSE)]:
$\text{MAE} = \left( \sum_{i=1}^{n} |y_i^p - y_i| \right) / n$         (11)
$\text{MRE} = \left( \sum_{i=1}^{n} |y_i^p - y_i| \right) / (y_i \times n)$         (12)
$\text{RMSE} = \sqrt{\left( \sum_{i=1}^{n} (y_i^p - y_i)^2 \right) / n}$         (13)
$\text{nRMSE} = \dfrac{\sqrt{\left( \sum_{i=1}^{n} (y_i^p - y_i)^2 \right) / n}}{\overline{y_i}}$         (14)
where $y_i^p$ is the predictive value of the different models, $y_i$ is the AGB of the $i$th sample plot, and $n$
is the number of training datasets.
We then used the calculated MAE, MRE, RMSE, and nRMSE to identify the optimal model.
**2.4.5 Robustness of combined models**
To evaluate the robustness of the combined machine learning and spatial statistical models, we selected
22 independent sample plots (see details in S1 and S3 of the Supplementary Material) and made
nondestructive measurements of each tree in July 2019. We repeated the workflow used for
constructing the plot-level model and evaluated the models. We then evaluated whether the combined





models produced higher accuracy than the plot-level models by using the accuracy-assessment indexes
(MAE, MRE, RMSE, and nRMSE).
**2.5 Model application and upscaling**
We treated the irregular polygon forest patches (2980 patches) of the Forest Management and Planning
Inventory (FMPI) as a homogenous sample plot and used the optimal plot-level model to upscale forest
AGB (see section S1 of the Supplementary Material). We then compared the upscaled forest AGB with
the AGB map obtained from the allometric model and calculated the MRE of AGB between the two
methods (see Eq. A.15 in section S1 of the Supplementary Material).
**3 Results**
**3.1 True AGB of sample plots**
The true AGB for the 30 sample plots ranged from 1.02 to 135.79 Mg·ha$^{-1}$, with an average value of
47.34 Mg·ha$^{-1}$ and a standard deviation of 34.46 Mg·ha$^{-1}$. The coefficients of variation of the AGB for
all the sample plots and for the 10 age categories were 0.73 and 0.07–0.37, respectively.
**3.2 Spatial distribution test and the selection of variables**
**3.2.1 The effect of different variables**
Figure 4 shows the correlation-coefficient matrix of variables. The following variables were strongly
correlated with AGB: longitude $(r = -0.56)$, DBH $(r = 0.79)$, H $(r = 0.84)$, trunk volume
$(r = 0.86)$, and forest age $(r = 0.82)$. Timber volume and stem volume were both estimated based on
H and DBH, so they were excluded as covariates for the AGB plot-level models. To summarize, four
variables (longitude, DBH, H, and forest age) were selected as covariates for the AGB plot-level
models of the *Eucalyptus* forest in the Nanjing region. Table B.4 in section S2 of the Supplementary
Material lists the statistical descriptions of these covariates and the AGB statistics for the 30 sample
plots.



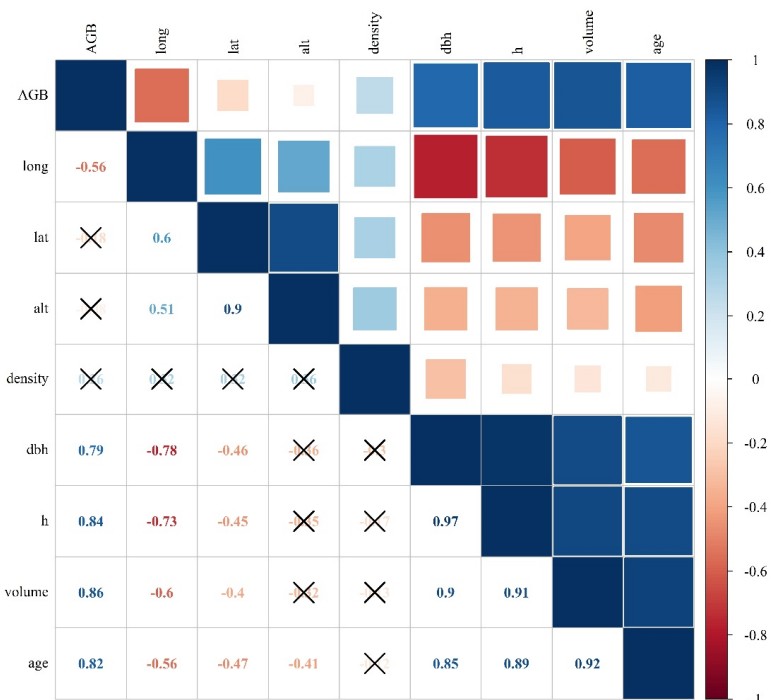



Figure 4. Pearson's correlation coefficients between AGB and other variables represented by numbers
and squares. Negative (red) numbers indicate that the corresponding variables are negatively correlated
and are colored in red, whereas positive (blue) numbers represent positive correlations. Larger absolute
numbers are indicated by darker colors, larger squares indicate stronger correlations, and the symbol "×

361                        " indicates insignificant correlations.

**3.2.2 Spatial autocorrelation test**
The spatial distribution of the true AGBs of the 30 sample plots displayed a pattern of aggregation (see
red regions in Fig. C.1, section S3 of the Supplementary Material and Table 1). In addition, because
less than 1% of the AGB data were randomly distributed (see blue regions in Figs. C.1 and S3 of the
Supplementary Material and Table 1), the possibility of an aggregated distribution was greater than that
of random distribution. Furthermore, the null hypothesis was significantly rejected ($p < 0.01$). These
results suggest that the spatial distribution of the AGB data displays aggregation and a pattern of strong




spatial autocorrelation.

370                          Table 1. Spatial autocorrelation and heterogeneity test.

| Spatial autocorrelation | | Spatial heterogeneity | | |
|---|---|---|---|---|
| **Items** | **Values** | **Factors** | **$q$ value** | **$p$ value** |
| Moran I | 0.36 | AGB | 0.87 | <0.01 |
| | | Longitude, long | 0.38 | <0.01 |
| $z$-score | 4.78 | Diameter at breast height, DBH | 0.54 | <0.01 |
| | | Tree height, H | 0.63 | <0.01 |
| $p$-value | 0.00 | Age | 0.92 | <0.01 |

**3.2.3 Spatial heterogeneity test**
As shown in Table 1, the true AGBs of the sample plots were divided into three strata by using $k$-means
clustering. We then ran the GeogDetector model and obtained a $q$ value of 0.87 and a $p$ value less
than 0.01. These results indicate that the within-layer variances were far less than the sum of variances
among different strata. The results also suggest that the reference AGBs of the 30 sample plots were
associated with obvious spatially stratified heterogeneity.
**3.3 Performance of plot-level models**
We developed seven models for estimating AGB: three machine learning models (SVM, RBF-ANN,
and RF), one spatial statistical model (P-BSHADE), and three combined models that integrated each
machine learning method with the spatial statistical method (SVM & P-BSHADE, RBF-ANN &
P-BSHADE, and RF & P-BSHADE). Furthermore, we used the leave-one-out cross-validation method
to split the datasets and evaluated the prediction performance of these seven methods based on the
indicators MAE [Fig. 5(a)], MRE [Fig. 5(b)], RMSE [Fig. 5(c)], and nRMSE [Fig. 5(d)].



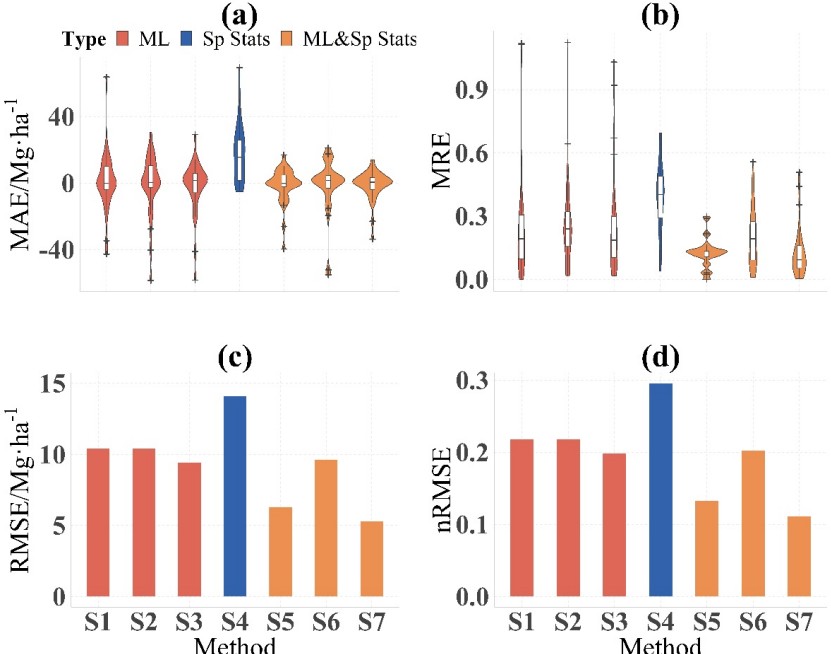

Figure 5. Prediction performance of the seven different models. (a) MAE and (b) MRE are presented as boxplots for each prediction method, with the median (black horizontal line in the box), inter-quartile range (25%–75% in the box), the range 5%–95% (whiskers), and outliers (asterisks) labeled (S1=SVM, S2=RBF-ANN, S3=RF, S4=P-BSHDE, S5=SVM & P-BSHDE, S6=RBF-ANN & P-BSHDE, S7=RF & P-BSHDE, ML=machine learning, Sp Stats=Spatial statistics). Histogram distributions of RMSE and nRMSE for each prediction method are presented in panels (c) and (d), respectively.

The forest AGB estimates obtained by the three machine learning methods were significantly more accurate than those obtained by the spatial statistical method. The performance indicators for P-BSHADE were MAE=18.37 Mg·ha$^{-1}$, MRE=39.13%, RMSE=14.08 Mg·ha$^{-1}$, and nRMSE=29.57%, whereas those for the machine learning methods covered the following ranges: MAE 10.16–12.15 Mg·ha$^{-1}$, MRE 24.79%–26.69%, RMSE 9.43–10.39 Mg·ha$^{-1}$, and nRMSE 19.80%–21.82%.

Among the three machine learning methods, the accuracy of RF was highest. The four evaluation indexes (MAE=10.16 Mg·ha$^{-1}$, MRE=25.93%, RMSE=9.43 Mg·ha$^{-1}$, and nRMSE=19.80%) were





substantially less than those for P-BSHADE and those for the other two machine learning methods
(MAE=11.17–12.15 Mg·ha$^{-1}$, MRE=24.79%–26.69%, RMSE=10.39–10.39 Mg·ha$^{-1}$, and nRMSE =
21.82%). Finally, the combination of machine learning and spatial statistical models produced smaller
MAE (5.68–10.14 Mg·ha$^{-1}$), MRE (12.47%–20.49%), RMSE (5.30–9.63 Mg·ha$^{-1}$), and nRMSE
(11.13%–20.23%) than the single machine learning methods. Of the three combined methods, RF &
P-BSHADE produced the highest accuracy with the smallest MAE (5.68 Mg·ha$^{-1}$), a modest MRE
(12.97%), and the smallest RMSE (5.30 Mg·ha$^{-1}$) and nRMSE (11.13%). In contrast, RBF-ANN &
P-BSHADE had the highest MAE (10.14 Mg·ha$^{-1}$), MRE (20.49%), RMSE (9.63 Mg·ha$^{-1}$), and
nRMSE (20.23%). Compared with the RF model, the RF&P-BSHADE model led to a reduction of the
cross-validated prediction error of 43.80%~50.00% (44.08% for MAE, 50.00% for MRE, and 43.80%
for RMSE and nRMSE).
We also explored the relationship between the observed and predicted AGBs in terms of
cross-validation results (Fig. 6). The quantity $R^2$ was calculated for the linear regression model applied
to the observed and predicted AGBs; $R^2$ for every model was greater than 0.9. Although P-BSHADE
had the highest $R^2$, its distribution of dots in Fig. 6(d) differed quite significantly from the 1:1 line. Of
the seven models, the accuracy of RF & P-BSHADE was the highest and the distribution of dots in Fig.
6(g) was closest to the 1:1 line. Therefore, we concluded that RF & P-BSHADE was the optimal
model.


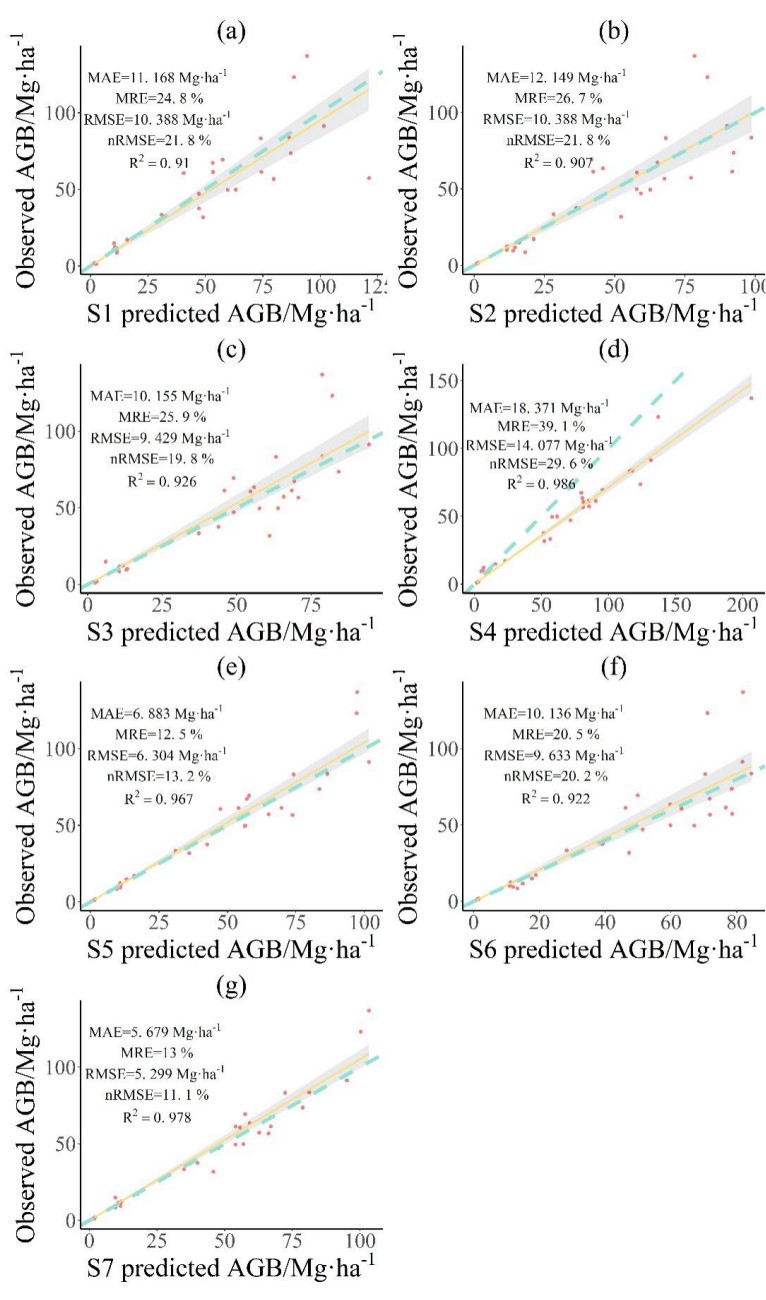

Figure 6. Comparisons of predicted and observed AGBs for accuracy assessment. Panels (a)–(g) show

SVM (S1), RBF-ANN (S2), RF (S3), P-BSHADE (S4), SVM & P-BSHADE (S5), RBF-ANN &

P-BSHADE (S6), RF & P-BSHADE (S7), respectively. Green dashed lines represent a 1:1 relationship;

dots represent individual sample plots; solid yellow lines indicate trend lines for dots.



We compared three machine learning methods with three corresponding combined machine learning
and spatial statistical methods by using differences in MAE, MRE, RMSE, and nRMSE during two
periods, 2012 and 2019 (Fig. 7). The results suggest that the combined models improved the accuracy
of single machine learning models during both years. This suggests that the combined methods are
robust.

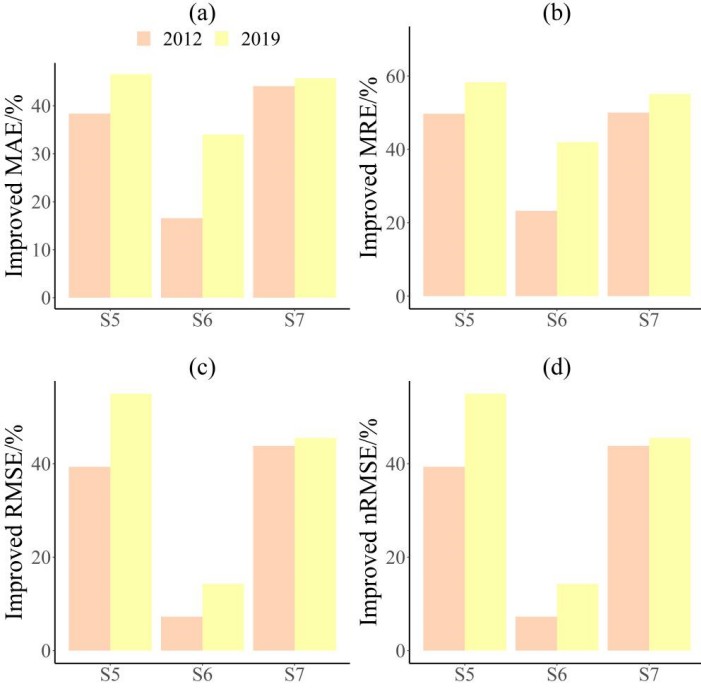


Figure 7. The improvement in accuracy assessment indexes of three combined machine learning and
spatial statistical methods by comparison with three corresponding machine learning methods. Panels
(a)–(d) show the MAE, MRE, RMSE, and nRMSE, respectively; S1-S5 represents RMSE comparison
of S5 with S1, S2-S6 represents RMSE comparison of S6 with S2, and S3-S7 represents RMSE
comparison of S7 with S3 (S1=SVM, S2=RBF-ANN , S3=RF, S4=P-BSHDE , S5=SVM & P-BSHDE,
S6=RBF-ANN & P-BSHDE, S7=RF & P-BSHDE).

Figure C.3 in section S3 of the Supplementary Material shows the spatial distribution of AGBs



predicted by the RF & P-BSHADE model. The predicted AGBs were 7.54–89.93 Mg·ha$^{-1}$, with an
average of 41.21 Mg·ha$^{-1}$, a median of 43.53 Mg·ha$^{-1}$, a standard deviation of 18.83 Mg·ha$^{-1}$, and a
coefficient of variation of 45.69%. The total AGB of the Nanjing region (2980 forest patches)
estimated by RF & P-BSHADE was 122 812.1 Mg, whereas that estimated by the allometric model
was 123 021.5 Mg. The percent difference in total AGB between the two methods was 0.17%.
Meanwhile, the AGB MRE between the two methods ranged from 0.04% to 99.8%, with an average of

443   19.93%.

**4 Discussion**

we developed, evaluated, and compared the accuracy and performance of three different machine
learning models [support vector machine (SVM), random forest (RF), and the radial basis function
artificial neural network (RBF-ANN)] in this study, which contains one spatial statistics model
(P-BSHADE) and three combinations thereof (SVM & P-BSHADE, RF & P-BSHADE, ANN &
P-BSHADE) on forest AGB estimates. Those findings suggested that the combined models, especially
the RF & P-BSHADE model, could improve the accuracy of plot-level AGB estimates and could reduce
the uncertainty of plot-level AGB estimates, owing to its integrated the theoretical advantages of
machine learning and spatial statistics.

**4.1 Significance of the optimal AGB model at the plot-level**

In the past, ecologists converted AGB estimates from forest sample plots into regional AGB estimates
by scaling up from the tree-level to the regional scale (Malhi et al., 2004). Plot-level AGB models
therefore link tree-level AGB models to regional-scale AGB models. Research by Chen et al. (2015)
found that ignoring the uncertainty of plot-level models increased the total uncertainty of pixel-level
estimates by 6%. In addition, Marvin et al. (2014) found that the distribution pattern of most AGB is
either non-Gaussian, skewed, or multi-modal, especially in tropical and subtropical regions. Different
intensity and direction of factors are coupled together, resulting in high heterogeneity and clear
nonlinearity in the spatial distribution of forest AGB.
Here, we integrated the advantages of machine learning and spatial statistics at the plot level (the key
scale linking the tree-level scale to the landscape scale) to construct a plot-level AGB model for a





subtropical region. The approach provides a high-precision plot-level AGB model whose estimates can
be compared with those obtained from remote sensing, ground observations, and model simulations. It
also provides a foundation for making informed forest management decisions (e.g., the method enables
quantitative evaluation of carbon emissions from deforestation). Combining the advantages of
machine-learning-based quantification of AGB and the complex nonlinear relationships between
multiple environmental covariates, in conjunction with the P-BSHADE model, allows the spatial
autocorrelation and heterogeneity of multiple environmental covariates to be incorporated into the model.
In addition, the sample points are subsequently rectified, thus leading to the best linear unbiased estimate
of the target plots.

### 473    4.2 Model comparisons

#### 474    4.2.1 Machine learning outperforms the spatial statistical model

Regarding the AGB plot-level models, the machine learning methods outperformed the spatial statistical
method (P-BSHADE) in terms of prediction accuracy. This may be because machine learning offers an
array of supervised learning models capable of relating forest AGB to multi-variables, including forest
variables and environmental variables, via complex, potentially nonlinear functional relationships.
Machine learning models appear adept at tackling high-dimensional problems, particularly in areas
where effective algorithms are lacking and where programs must dynamically adapt to changing
conditions (Görgens et al., 2015; Latifi et al., 2010; Stojanova et al., 2010). In addition, the P-BSHADE
model yielded negative weights between a small number of plots, which might introduce a slight degree
of uncertainty into the results (Xu et al., 2013). Our results were consistent with those of Povak et al.
(2014) and Li et al. (2011), who found that a machine learning method (RF) outperformed the spatial
statistical method (e.g., Geographically Weighted Regression, Inverse Distance Weighting ) in terms of
prediction accuracy.

#### 487    4.2.2 Why a combined model outperforms a single machine learning or spatial statistical model

As expected, the prediction accuracies of the combined methods were higher than those of any single
method (either machine learning or spatial statistical). This may due to the advantages of machine
learning, which can compensate for the inherent defects of the P-BSHADE model, and vice versa.
On the one hand, the P-BSHADE model has its own merits: (1) It takes into account the spatial



autocorrelation and spatial heterogeneity of the distribution of the target objects, not only to solve the
difference between target objects caused by the different terrain or geographical location but also to
solve the problem of strong correlation between target objects with remote geographical locations due
to similar terrain condition. (2) The P-BSHADE model calculates the covariance between objects by
using a reference sequence between objects (which means the reference AGB data between plots in our
study). This method is more reliable because it avoids the second-order stationary hypothesis (i.e.,
when using the Kriging algorithm, semi-variograms need this hypothesis), which does not correspond
with the actual situation. (3) P-BSHADE regards strongly correlated plots as neighboring plots.
However, the P-BSHADE model is also handicapped by the fact that the founding assumption does not
conform to reality. The assumption is that estimated AGB is accurate in all sampling plots except the
target sampling plot. In other words, the premise behind using only the P-BSHADE model is that the
reference AGB data is accurate or strongly correlated with AGB. In reality, the AGB of each sampling
plot has a varying degree of uncertainty because it is obtained from the allometric model. Since the
P-BSHADE model combined with machine learning uses the results optimized by machine learning as
the reference series, it further improves the accuracy of AGB mapping.
Machine learning also has its advantages and disadvantages. As we described in the previous section
(4.2.2), machine learning has the advantage of being able to handle complex, potentially nonlinear
relationships between forest AGB and other variables. However, the initial samples of machine
learning are randomly selected, which may lead to differences in the results of each operation of the
model. In addition, machine learning uses the average value of all regression trees in the calculation,
which may result in overestimating the lower value and underestimating the higher value. As opposed
to machine learning, the P-BSHADE model takes into account the spatial autocorrelation and spatial
heterogeneity of forest AGB and of environmental covariates, and the bias of the observed values of the
sampling plots, which corresponds more to actual situations. A combined model takes the result of
machine learning as the reference series of P-BSHADE, so that the fitting process of the combined
model takes spatial relationships more into account than is the case for the single machine learning
model. The end result is improved accuracy.
Machine learning models or the P-BSHADE model have been used to model the uncertainty of
temperature measurements obtained by weather stations (Fassnacht et al., 2014; Paul et al., 2016; Xu et



al., 2013). However, the methods used in these studies were adopted independently. Conversely, the
combination of machine learning and spatial statistics can improve the prediction accuracy of AGB
maps, which in turn can be used as criteria for improving the accuracy of LiDAR remote-sensing
technology and the results of ecological process models. Eventually, these improvements can promote
process-oriented projects that require dynamic AGB predictions for large-scale forests in different
forest management scenarios.
In addition, we compared the prediction accuracy of AGB mapping obtained by the combined spatial
statistical and machine learning models with that reported by recent studies using AGB plot-level
models. In the current literature on remote-sensing estimation of forest AGB, nRMSE, RMSE, and $R^2$
were commonly used as indexes for evaluating the prediction performance of models affected by
research sample size, data type, and forecasting methods (Fassnacht et al., 2014). In contrast, the
present study used four conventional indexes for evaluating prediction performance: nRMSE, RMSE,
MAE, and MRE. The criterion for model selection is to choose indexes summarized from sample
prediction (such as nRMSE), rather than choosing the goodness-of-fit $R^2$ (Babcock et al., 2015). Based
on calculated nRMSE indexes, the AGB prediction accuracy of the combined RF & P-BSHADE model
(11.13%) was higher than that obtained by Babcock et al. (2015) (33.91%) in Colorado, USA. In that
study, the authors used a combination of airborne LiDAR, a forest inventory database, and a Bayesian
spatial hierarchical framework model and introduced spatial random effects to compensate for the
residual spatial dependence and non-stationary model covariates. The AGB prediction accuracy of the
method developed in the current work was also greater than that obtained by Ioki et al. (2014)
(nRMSE=26%) in northern Borneo using a stepwise linear regression model with airborne LiDAR and
a ground survey. Furthermore, it exceeded the accuracy obtained by Hansen et al. (2015) in the tropical
submontane rain forest (34.4%) using fusion maps of multi-source databases combined with multiple
regression analysis. Our prediction accuracy is close to that obtained by Kim et al. (2016) (9.2%) who
studied an intact tropical rain forest by using a voxel-based method based on airborne LiDAR in
conjunction with field monitoring in Brunei. Our combined methods produce very small RMSE for the
prediction accuracy of AGB, which we attribute to the following reasons: (1) The true AGBs of the 30
sample plots were calculated from each tree by using an allometric model constructed from the 90 most
accurate harvested trees. There were no differences in the range of true values. (2) Machine learning



methods were used to quantify the complex nonlinear relationship between AGB and multiple
environmental covariates. (3) We applied a spatial statistical method based on the hypothesis of spatial
heterogeneity. Although the nRMSE index was calculated by different studies using different datasets
and prediction methods in different locations, most studies agreed that nRMSE was the most
commonly used indicator for measuring the AGB prediction errors of plot-level models and for
calculating the true AGB of forest sample plots. In contrast to other studies, our work reflects not only
a focus on subtropical forests but also the methodological differences in uncertainty mitigation,
especially in terms of comprehensively addressing the sources of uncertainty caused by multiple spatial
and environmental covariates.
**4.2.3 Why RF & P-BSHADE method outperforms other combined methods**
The three combined machine learning and spatial statistical methods produced more accurate AGB
predictions than any individual method. The accuracy of the RF & P-BSHADE and SVM &
P-BSHADE methods were significantly higher than that of the individual methods, but the RBF-ANN
& P-BSHADE method was only slightly higher. The accuracies of the combined methods depend on
the accuracy of the reference series (machine learning predicted result) (Xu et al., 2013). In other words,
the higher the accuracy of the predicted machine learning results, the higher the accuracy of the
combined method. Therefore, the different improvements offered by the three combined methods may
be attributed to the following two mechanisms: (1) the RF and SVM models are easier to use and
optimize than RBF-ANN (Raczko and Zagajewski, 2017). RBF-ANN is sensitive to hyper-parameters
and usually requires optimized parameters to obtain better fitting results. However, in the present study,
we used no optimized algorithms, such as genetic algorithms, to obtain parameters in the machine
learning model. Furthermore, the number of training samples determines the number of nodes in the
hidden layer of the RBF-ANN model, and the number of nodes significantly affects the prediction
accuracy. With only 30 training samples used in this study, the combined approach may have been
unable to strongly improve prediction accuracy. (2) RBF-ANN is more suitable for nonlinear stochastic
dynamic systems (Elanayar and Shin, 1994), whereas the relationship between AGB and environmental
covariates in this study is likely a monotonically increasing function.



### 4.3 Comparing upscaling of RF&P-BSHADE with allometric model

We used FMPI data to upscale the optimal plot-level AGB model from plot level to region scale. Because the allometric model offers a fast and simple calculation method, it has been used in many studies as the basis for determining the benchmark map. Nevertheless, spatial heterogeneity caused by multiple environmental covariates is not considered in the allometric model because potential errors in the AGB estimate may be propagated and affect the accuracy of the regional AGB map. Although we regarded the FMPI patches as homogeneous study units in the present study, the area of the forest patches is significantly larger than that of the sample plots. Upscaling results will thus have large uncertainties (see Figs. C.4, S3 of Supplementary Material) (Chen et al., 2015). The current study finds that the relative percent difference in total AGB between RF & P-BSHADE and the allometric model was 0.17%. Meanwhile, the relative error (RE) in AGB between the two models ranged from 0.04% to 99.8% with a MRE of 19.93%. This suggests that the two methods are similar in terms of overall estimates of AGB but that the local spatial distribution of AGB differs. Differences in AGB spatial distribution have been reported in many studies of AGB maps. Babcock et al. (2015) asserted that the main reasons for the differences in the spatial distribution of AGB maps between different methods include the following: (1) The structural framework of different research methods and schemes cannot truly reflect actual forest growth. (2) The model is usually a simplification of an ecological process and ignores spatial heterogeneity at the regional scale. (3) The model does not consider the influence of multiple environmental covariates (vegetation, topography, and others) on forest growth in the region.

### 5 Conclusions

This paper proposes a method to integrate the advantages of machine learning and spatial statistics, different datasets, and multiple environmental covariates to improve the accuracy of plot-level AGB-estimation models. In this study, we explored the prediction performance of different AGB models and found that the model that combines the Random Forest and P-BSHADE models substantially improved estimates of forest AGB. Although data from the sample plots and harvested trees were collected only from *Eucalyptus* forests in the Nanjing region of China, the proposed model and the associated results can provide references for AGB mapping in other countries and in different types of tropical forests.





**Data availability.**
All data are included in the paper and Supplement.
**Author Contributions**
Y.R. designed the study. X.Z. carried out the data collection. S.D. carried out the analyses and
visualized the data. X.Z. and S.D. wrote the manuscript with help from Y.R. L.G., C.X., S.Z., Q.C., and
X.W. provided technical advice and guidance throughout the project implementation and paper-writing
stages. S.D. and X.Z. contributed equally to this work.
**Competing interests.**
The authors declare that they have no conflict of interest.
**Acknowledgments**
Shaoqing Dai and Xiaoman Zheng contributed equally to this work and should be considered as
co-lead authors. This work was supported by National Science Foundation of China (Grants No.
31670645, No. 31972951, No. 31470578, No. 31200363, No. 41801182, and No. 41807502), National
Social Science Fund (17ZDA058), the National Key Research Program of China (2016YFC0502704),
Fujian Provincial Department of S&T Project (Grants No. 2016T3032, No. 2016T3037, No.
2016Y0083, No. 2018T3018, No. 2019J01136, and No. 2015Y0083), the Strategic Priority Research
Program of Chinese Academy of Sciences (XDA23020502), the Ningbo Municipal Department of
Science and Technology(2009C10056), the Xiamen Municipal Department of Science and Technology
(Grants No. 3502Z20130037 and No. 3502Z20142016), the Key Laboratory of Urban Environment and
Health of CAS (KLUEH-C-201701), the Key Program of the Chinese Academy of Sciences
(KFZDSW-324), and the Youth Innovation Promotion Association CAS (2014267). We are grateful to
the anonymous reviewers for their constructive suggestions.




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
