# Peer review of "Improving maps of forest aboveground biomass: A combined approach using machine learning with a spatial statistical model"

_Biogeosciences, 2020_

## Short Comment (SC1) · 7 Mar 2020

This manuscript attempts to improve AGB maps by combining the machine learning model (SVM, RF, and RBF-ANN) and spatial statistical model (P-BSHADE). Overall the manuscript seems technically sound and, in most cases, is well written. The experiment is designed for one type of forest (Eucalyptus forest), thus I'm afraid the influence is limited. The results are reasonable, I question a number of aspects of the source data. Based on the comments below I suggest major revisions.

General questions: Q1: Your number of sample plots (N=30) is too small for machine learning models. How could the sample plots represent the region? The range of

biomass (1.02 – 135.79 Mg/ha). Did this range cover the whole range of your target species? The age of harvest trees ranged from 1 to 10 yr. What's the age range in the study area? What's the DBH range of your 90 harvested trees? Same questions as biomass and age.

Q2: How did you combined machine learning and spatial models? I got confused after reading your descriptions. Comparing to other methods, your description of the P-BSHADE model a little bit lengthy, suggest moving details into the supplement.

Q3: Your results suggest that plot-level biomass models need to be built per species and per ecoregion? The problem of not using allometric models is, how to quantify the AGB of not-so-common species?

Q4: Did you consider the influence of plot size? Say could your model build using a 20-m plot applied to 40-m or 100-m scale? This important when considering the need to apply models at larger geographical domains via the combined use of remote sensing datasets.

Q5: This study constructs local AGB allometric models, for a small Eucalyptus forest in Nanjing county. However, how should we apply your method in other places over a large geographical domain?

Q6: Did you compare your models and existing allometric models within the region? What's the influence of excluding small stems (living stem <8 cm) in your estimation of AGB?

Specific comments: L49: "the use of inadequate sampling data to construct the plot level prediction models" did you solve this issue?

L59: Selection of the allometric model could account for 20% uncertainty (Duncanson et al. 2017) Duncanson, L., Huang, W., Johnson, K., Swatantran, A., McRoberts, R., & Dubayah, R. (2017). Implications of allometric model selection for county-level biomass mapping. Carbon Balance and Management, 12

L82: Some recent studies integrated ground-based plot and remote sensing data for AGB mapping (Sun et al. 2011; Huang et al. 2019; Qi et al. 2019) Sun, G., Ranson, K.J., Guo, Z., Zhang, Z., Montesano, P., & Kimes, D. (2011). Forest biomass mapping from lidar and radar synergies. Remote Sensing of Environment, 115, 2906-2916

Huang, W., Dolan, K., Swatantran, A., Johnson, K., Tang, H., O'Neil-Dunne, J., Dubayah, R., & Hurtt, G. (2019). High-resolution mapping of aboveground biomass for forest carbon monitoring system in the Tri-State region of Maryland, Pennsylvania and Delaware, USA. Environmental Research Letters, 14, 095002

Qi, W., Saarela, S., Armston, J., Ståhl, G., & Dubayah, R. (2019). Forest biomass estimation over three distinct forest types using TanDEM-X InSAR data and simulated GEDI lidar data. Remote Sensing of Environment, 232, 111283

L85: "First, existing studies that used machine learning methods have not considered the spatial heterogeneity of multiple environmental covariates (such as longitude, latitude, and forest structure)" This statement is too arbitrary. What does "structure" refer to? Shouldn't structure information came from lidar or radar?

L96: "multiple environmental covariates (such as longitude, latitude, and forest structure)" A duplicate statement, modify to be concise;

L140: Suggest add equations of the allometric models you used here.

L175: What software/package did you applied to construct your model?

L179: (reference series)?

Figure 3. (b) SVM and (c) RF are for classification, not regression;

L445: "we" should be "We".

---

## Referee Comment (RC1) · Anonymous Referee #1 · 25 Mar 2020

The study presented in this manuscript compares the suitability of machine learning as well as geostatistical modelling approaches to retrieve maps of tree AGB on a regional scale based on plot-based surveys. The paper is reasonably written, however it is lacking a lot of important details that will be outlined in the more specific comments below. Apart from the need to improve the writing there is a clear need to improve the methodology.

GENERAL COMMENTS

1) Important information on the methodology are not given. In the method section it's not even mentioned which predictors were used for the machine learning model

training. But this essential as the success of the models is only marginally depending on the choice of the algorithm but in the first place on the ability of the variables being used to serve as predictors for AGB! Only in the result section we get an idea on the variables (longitude, DBH, H, and forest age). I'm surprised about these variables as remote sensing information (especially NDVI) would present much more obvious predictors for AGB when the aim is to model AGB on a regional scale. With the selected variables, how could you upscale the results to a regional scale? I guess neither DBH nor H are not available in a spatial continuous way. So your model cannot be used for regional mapping! However, your motivation is to use it for regional modelling so my question is a) can you really do it with your approach and b) if yes, why are you not doing so and also show the results in the manuscript?

2) The cross-validation strategy that you used is not suitable if you have spatially clustered data (as you obviously have looking at the map). This is shown by several studies (see references below, to mention just a few). What would be appropriate is a spatial cross-validation that is testing the ability of your model to make predictions for spatially new samples. At least you should take care that you never use data points from the same forest patch for both training and testing. Otherwise, it is not possible to evaluate the ability of your models for regional mapping. Including coordinates as predictors when the data are spatially clustered is very dangerous (see Meyer et al 2019) and can lead to high overfitting which can only be revealed with spatial cross-validation. So I recommend that in addition to spatial cross-validation, to perform a spatial variable selection (i.e. can the predictors be used to make predictions for new locations?)

Meyer, H., Reudenbach, C., Wöllauer, S., Nauss, T., 2019. Importance of spatial predictor variable selection in machine learning applications - Moving from data reproduction to spatial prediction. Ecological Modelling. 411, 108815.

Pohjankukka, J., Pahikkala, T., Nevalainen, P., Heikkonen, J., 2017. Estimating the prediction performance of spatial models via spatial k-fold cross validation. Int. J. Geogr. Inform. Sci. 31, 2001–2019. https://doi.org/10.1080/13658816.2017. 1346255.

Roberts, D.R., Bahn, V., Ciuti, S., Boyce, M.S., Elith, J., Guillera-Arroita, G., Hauenstein, S., Lahoz-Monfort, J.J., Schröder, B., Thuiller, W., Warton, D.I., Wintle, B.A., Hartig, F., Dormann, C.F., 2017. Cross-validation strategies for data with temporal, spatial, hierarchical, or phylogenetic structure. Ecography.

Valavi, R., Elith, J., Lahoz-Monfort, J.J., Guillera-Arroita, G., 2018. blockcv: an r package for generating spatially or environmentally separated folds for k-fold cross-validation of species distribution models. BioRxiv.

3) The concerns outlined above can be improved, however, I have also doubts about the general value of the paper. Relying on 30 plots only is very very limited for machine learning application (the spatial CV will probably reveal this). So I doubt that the results will produce results that allow for general conclusions on the value of combining machine learning with geostatistical modelling.

SPECIFIC COMMENTS

Line 25: I disagree that longitutde and latitude on this scale affect AGB. Even on a large scale they don't but are just proxies for e.g. climate but they are certainly not underlying factors for AGB on your small study area.

Line 49-51: One important thing is missing: The model might also fail because the predictor variables are not sufficient to estimate AGB.

Line 54: "An estimated 18%–103% of the uncertainty in AGB mapping can be attributed to model-dependent uncertainty". In fact between nearly nothing (∼18) and everything (>100). That sounds unreasonable, consider taking that sentence out.

Line 60-62: This differentiation between allometric models and statistical models does not seem to make sense. E.g. Allometric models can be based on linear relationships as well. Please improve the logical structure here.

Line 67-71: Be careful with the logical structure here as well: The major advantage is that machine learning is able to fit complex relationships which e.g. linear models

don't. And THEREFORE they might be advantageous in predictions (not "in addition" as you write in Line 72).

Line 74-81: The fundamental difference between the approaches is not getting clear here but this is important because combining the two approaches is the objective of the paper. In contrast to the statistical (including machine learning) approaches explained above, the spatial statistical approaches have the major assumption that "near things are more related that distant things". I think the general idea should be made clear and it should be explained why you expect that a combination might be the way forward.

Line 85-88 "studies that used machine learning methods have not considered the spatial heterogeneity of multiple environmental covariates (such as longitude, latitude, and forest structure)". I disagree. Most approaches use environmental covariates which of course have been heterogeneous as well.

No information on model tuning is given. Also please state which software implementations and settings of the algorithms you used.

Line 157-158: please explain why you tested for spatial autocorrelation etc. Why is this information relevant for the modelling?

Line 205:"Because of the Law of Large Numbers, RF does not overfit." That's wrong! Maybe random forest is robust to overfitting in terms of hyperparameter selection but it is not the case if you have data that are not independent. See e.g. the references mentioned above.

Line 206-207: Accurate predictions of random forest do NOT in the first place originate from injecting randomness. E.g. If the predictors are not sufficient to estimate a response variable, random forest will fail (and so will other algorithms)!

Fig. 6 : Is this based on the cros-validation?

Line 502:503: "The assumption is that estimated AGB is accurate in all sampling plots except the target samplig plot. In other words, the premise behind using only the P-

BSHADE model is that the reference AGB data is accurate or strongly correlated with AGB. " I don't understand that. The same reference data were used for all modedliung approaches and for sure we assume that the reference data are accurate for both types of models.

Line 578: "We used FMPI data to upscale the optimal plot-level AGB model from plot level to region scale." Did you? We don't get to see the results for the regional upscaling.

I wonder: Is your model really better than simply using the average measured AGB from each forest site as estimate for AGB for the entire patch?

---

## Author Comment (AC1) · 21 Apr 2020

The comment was uploaded in the form of a supplement:
https://www.biogeosciences-discuss.net/bg-2020-36/bg-2020-36-AC1-supplement.pdf

---

## Author Comment (AC2) · 21 Apr 2020

This manuscript attempts to improve AGB maps by combining the machine learning model (SVM, RF, and RBF-ANN) and spatial statistical model (P-BSHADE). Overall the manuscript seems technically sound and, in most cases, is well written. The experiment is designed for one type of forest (Eucalyptus forest), thus I'm afraid the influence is limited. The results are reasonable, I question a number of aspects of the source data. Based on the comments below I suggest major revisions.

**Response:**

Thank you for your consideration of our manuscript.

We agree that the articles published in *Biogeosciences* should be universal and representative, and the research results can help to solve the hot issues in ecology and geosciences. Therefore, we understand your concern about the influence of our manuscript. We are sorry that we did not clearly explain the innovation and influence of our manuscript. We explain this as follows:

1. The innovation of our manuscript is to integrate machine learning and a spatial statistical model. The integration of these two can help to complement each other 's advantages and improve the accuracy of the AGB estimation model. Machine learning has the advantage of being able to handle complex and potentially nonlinear relationships between forest AGB and other variables. However, the initial samples of machine learning were randomly selected, which may lead to differences in the results of each operation of the model. Additionally, specific machine learning algorithms have their own disadvantages, such as RF uses the average value of all regression trees in the calculation, which may result in the overestimation of the lower value and the underestimation of the higher value. As opposed to machine learning, the P-BSHADE model (a spatial statistical model) takes into account the spatial autocorrelation and spatial heterogeneity of forest AGB and of environmental covariates and remedies the bias of the observed values of the sampling plots in theory, which corresponds more to actual situations. A combined model takes the result of machine learning as the reference data (input data) of P-BSHADE so that the fitting process of the combined model accounts more for spatial relationships than is the case for the single machine learning model. In addition to the theoretical advantages of these methods, case studies presented in    this study also demonstrate the empirical superiority of the combined

model.

2. The allometric model is a simple, fast, and universal equation that is used in many studies. However, selection error in plot-level allometric modeling still leads to over 40% uncertainty (Djomo et al., 2016; Fayolle et al., 2013; Chave et al., 2014), and simple or complex forms of the allometric model account for 20% – 60% of the uncertainty (Picard et al., 2015). In our manuscript, we propose an improved method of AGB estimation which combines machine learning which is good at prediction with spatial statistical model which is good at reflecting spatial relationship to improve the estimation accuracy of the AGB model at the plot level.

3. Over the past 20 years, with the growing area of *Eucalyptus* plantations around the world, Brazil, India, China, Chile, Spain, DR Congo, Australia, South Africa, and other countries have established contiguous *Eucalyptus* planting areas. There have been many studies and reports on the study of the biomass estimation of *Eucalyptus* plantations combined with ecological process models at different temporal and spatial scales. China is the country with the largest area of planted forest in the world. However, China's planted forest suffers from the three practical problems of low productivity, unsustainability, and incongruous production function and ecological function, which urgently need to be solved by appropriate management measures. On the one hand, with the rapid growth of population, the timber demand is also increasing rapidly. On the other hand, there is the severe reality that total global forest resources have declined sharply in recent years. Thus, many countries and regions are vigorously developing fast-growing non-native trees to alleviate the contradiction between the supply and demand of timber and forest products in order to maintain economic and social development. *Eucalyptus* is one of the fastest-growing trees, and is controversial in the development of planted forest. A special study on *Eucalyptus* is of great significance.

4. Although *Eucalyptus* is taken as a case study in our manuscript, the model we proposed can also be applied to other tree species or mixtures of tree species. There is no particular relationship between our model settings and tree species. No unique characteristics of *Eucalyptus* are added to the model, and other forest types also can provide the input data such as tree height, DBH, longitude, and other variables. Therefore, the model we proposed can be expanded and its

influence is not limited to *Eucalyptus* forests. In fact, this idea of improving the accuracy of the model also has the potential to solve other ecological problems.

To sum up, we did not clearly express our meaning at first, however, in view of the influence of the four aspects of the manuscript we described above, we believe it can attract attention and citation. I hope it has been explained clearly now.

**General questions:**

**Q1:** Your number of sample plots (N=30) is too small for machine learning models. How could the sample plots represent the region? The range of biomass (1.02 – 135.79 Mg/ha). Did this range cover the whole range of your target species? The age of harvest trees ranged from 1 to 10 yr. What's the age range in the study area? What's the DBH range of your 90 harvested trees? Same questions as biomass and age.

**Response to Q1:**

1. For machine learning, the data volume of 30 plots is small. In practice, it is very difficult to obtain 30 plots in forestry, in which three trees were cut down for each plot, tree height and DBH of each tree were measured for each plot to obtain the most real plot data. If conditions permit, we attempt to obtain more representative plot data. However, we must also consider the cost–benefit problem in practical applications. Therefore, we took three measures to make up for the possible problems caused by the small sample size. The first was to use the spatial block cross-validation to make full use of the existing samples, and also to avoid overfitting; the second was to combine the P-BSHADE spatial statistical model to remedy the fitting results biased in machine learning; and the third was to investigate another 22 sample plots to test our proposed model again.

2. The setup of 30 plots takes into account the spatial heterogeneity of the whole study area and was carried out strictly according to the selection criteria of a set of representative forestry plots. We have included a detailed introduction of how the 30 sample plots were determined in the Supplementary Material as follows:

*"The purpose of plot selection was to establish fixed and permanent plots representing regional Eucalyptus growing conditions and to provide harvested tree data on the single-tree scale with adequate consideration of spatial heterogeneity. Patches were selected first and met the following six conditions: (1) patch records were available from FMPI data for 2009; (2) forest stands were classified as timber or commercial forest; (3) forest patches were disturbance-free during the previous seven years, including but not limited to logging, fire, and pests; (4) forest patches were not replanted; (5) patches contained closed canopy forests; and (6) patches were monocultures, not mixed stands. Based on these six conditions, 2,980 Eucalyptus patches were selected from the FMPI data and fixed and permanent plots were established. The 2,980 selected patches were divided into ten groups based on forest age. Each stand group had been planted at the same time. We calculated the mean basal area for each group and used this as the basis for fixed plot selection, which was obtained from specified plot design and sampling procedures. In parallel, we considered site conditions, forest use, and forest origin (natural vs. man-made), and subsequently established 30 permanent square plots (20 m × 20 m)."*

3. The AGB of the 30 sample plots was found to be 1.02–135.79 mg/ha, which covers the regional AGB of 4.70–119.78 mg/ha calculated with the allometric model and the regional AGB of 7.54–89.93 mg/ha calculated with the optimal model (RF and P-BSHADE method).

4. The age range of *Eucalyptus* forest in the study area is 1–51 years, however only 24 out of the 2980 patches are over 10 years old. According to the growth habits and characteristics of *Eucalyptus*, its growth after the age of 10 is very slow compared with that in the previous 10 years. In China, in order to maximize economic benefits, the common practice is to cut down *Eucalyptus* trees that have been growing for 10 years for commercial use and replant new seedlings. The 24 of these small patches may not be cut down for special reasons.

5. The DBH in the 30 sample plots ranged from 2.1–18.4 cm, and that in the study area ranged from 5.0–60.0 cm. Of the 2980 patches, only 53 had DBH values larger than 18.4 cm, 22 had values larger than 20 cm, and only two had values larger than 40 cm. It should be noted that 90 trees were selected and cut down based on the average DBH of each plot, so the DBH of the 90 trees were respectively at the average level of their forest age.

**Q2:** How did you combined machine learning and spatial models? I got confused after reading your descriptions. Comparing to other methods, your description of the P-BSHADE model a little bit lengthy, suggest moving details into the supplement.

**Response to Q2:**

P-BSHADE is a spatial statistical inference model. It requires the reference value of the target object (which is forest AGB in our manuscript) as input data when estimating. Determining how to obtain the reference value is the key to combining the P-BSHADE model with other methods. In our manuscript, we first used a machine learning method to estimate AGB and then used this estimate as a reference value to input into the P-BSHADE model to estimate AGB again.

The revised figure is as follows:

Also, thank you for your suggestion, we have moved details of P-BSHADE into the Supplementary Material (Section S1).

[Figure]

Q3: Your results suggest that plot-level biomass models need to be built per species and per ecoregion? The problem of not using allometric models is, how to quantify the AGB of not-so-common species?

**Response to Q3:**

We suggest to determine whether it is necessary to establish a plot-level model for per species and per ecoregion according to service objectives, cost-effectiveness, etc.

Allometric models has been widely used to estimate forest biomass, however biogeographic variation in allometric relations, wood density, and soil fertility introduce sources of uncertainty,

which will be expanded and propagated in the next step when the model is applied to the region. The combination of machine learning and spatial statistics proposed in our study can reduce the error caused by the plot-level model. In our manuscript, although we take pure *Eucalyptus* forest as a case study, it does not mean that our approach can only be used to establish a plot-level model for *Eucalyptus*. Rather, our model can be applied to both pure species forests and mixed species forests, and furthermore can be applied to a single ecoregion or a larger-scale ecoregion, as long as enough sample plots representing the research region can be collected. When a high-precision biomass map with clear spatial distribution is required, our model can help to reduce the error of the plot-level model. This method is attractive as the plot-level model allows the subsequent construction of large-scale regional biomass maps.

**Q4:** Did you consider the influence of plot size? Say could your model build using a 20-m plot applied to 40-m or 100-m scale? This important when considering the need to apply models at larger geographical domains via the combined use of remote sensing datasets.

**Response to Q4:**

Yes, the influence of plot size is very important. In order to avoid this influence, the model established in our manuscript can only be applied to 20-m plots. Considering the universality of the model and the convenience of direct comparison between different research results, a 20-m plot size is usually selected in subtropical regions. If it is necessary to apply the model to plots with a size of 40-m or 100-m, it is better to establish plot-level models optimized for these sizes. That is, the setting of the model plot size should be adjusted according to the resolution of the data to be combined in the next step when the model is to be applied to the regional scale.

The resolution of remote sensing data and plot size are very important for up-scaling in the estimation of forest biomass. In particular, the sample plots not only provide validation data for large-scale prediction but also provide multi-source environmental variable information that can be used to calibrate models. Therefore, it is very important to set the plot size to be consistent with the resolution of the remote sensing data to be combined in the next step. However, the actual situation

is that the larger the plot size, on the one hand, the required manpower and financial investment will increase, and on the other hand, the increase of workload and difficulty may affect the accuracy of the sample data. There have been many studies on the impact of sampling schemes (such as different sample sizes and plot sizes), model selection (such as different model types), and data selection (such as different types of remote sensing data) on the accuracy of forest biomass estimation and on how to organically combine the above three factors to obtain the optimal scheme. We think that this is another important issue, however it is beyond the scope of our manuscript.

**Q5:** This study constructs local AGB allometric models, for a small Eucalyptus forest in Nanjing county. However, how should we apply your method in other places over a large geographical domain?

**Response to Q5:**

We want to share and promote our model which combines machine learning and spatial statistics and thus combines the advantages of both in order to improve the accuracy of AGB estimation. Although in our study we took pure *Eucalyptus* forest as a case study, this model is not limited to the study of *Eucalyptus* forest, and furthermore can also be applied to other regions, including larger geographical areas (such as China, Asia, etc.). The application of this method to a larger geographical area should also follow the up-scaling steps from tree-level to plot-level to regional level. The most significant contribution of our model is to reduce the estimation uncertainty at the plot level, reduce the extent of further propagation or up-scaling of the uncertainty, and prepare for the combination of plot-level results and remote sensing data to realize the up-scaling of estimation and obtain high-precision distribution maps in the future.

**Q6:** Did you compare your models and existing allometric models within the region? What's the influence of excluding small stems (living stem <8 cm) in your estimation of AGB?

**Response to Q6:**

1. Our team has previously established plot allometric models of *Eucalyptus* forest biomass within this region (Qiu et al., 2018), however, we did not produce AGB allometric models for this region then. But in this manuscript, we have already established an AGB allometric model of the sample plots within this region. When using the P-BSHADE model, we used the results of the allometric model as reference AGB data. An accuracy comparison with other models established in our manuscript is shown in the table below:

|   | Method | MAE | MRE | RMSE |
|---|--------|-----|-----|------|
| 1 | SVM | 11.17 | 0.25 | 10.39 |
| 2 | ANN-RBF | 12.15 | 0.27 | 10.39 |
| 3 | RF | 10.16 | 0.26 | 9.43 |
| 4 | P-BSHADE | 18.37 | 0.39 | 14.08 |
| 5 | SVM&P-BSHADE | 6.88 | 0.12 | 6.30 |
| 6 | ANN-RBF&P-BSHADE | 10.14 | 0.20 | 9.63 |
| 7 | RF&P-BSHADE | 5.68 | 0.13 | 5.30 |
| 8 | Allometric Model | 14.90 | 0.53 | 23.04 |

2. After re-examining the data processing, we found that trees with DBH values of less than 8 cm were also included. At the beginning, the data collecting plan excluded trees with DBH values of less than 5 cm according to the "Detailed rules for the Eighth National Forest Resources Inventory of Fujian Province" standard. However, in the practical investigation, we found that most *Eucalyptus* forests aged 1−3 years had DBH values of less than 8 cm. If we had followed the aforementioned standard, our experiment would not have been carried out. Therefore, the actual operation later included all the trees, no matter what their DBH were. We are sorry for the mistake in the manuscript.

L49: "the use of inadequate sampling data to construct the plot level prediction models" did you solve this issue?

**Response:**

We did not resolve this issue, since it involves sampling design. Our manuscript points out two primary sources of uncertainty in regional biomass maps, and we also mentioned that the second source of uncertainty (L52-53) was the main issue we were attempting to solve in this manuscript, as follows. Please check this. Although the first source falls outside the scope of this manuscript, we will study it in the future.

*"The uncertainty of such regional maps can be attributed to two primary sources: (1) the use of inadequate sampling data to construct the plot level prediction models, and (2) model-dependent uncertainty, including unreasonable model-parameter assumptions and improper model structure (Chen et al., 2015; Gao et al., 2016; McRoberts et al., 2016). The present study mainly focuses on reducing the second source of uncertainty."*

**L59:** Selection of the allometric model could account for 20% uncertainty (Duncanson et al. 2017)

Duncanson, L., Huang, W., Johnson, K., Swatantran, A., McRoberts, R., & Dubayah, R. (2017). Implications of allometric model selection for county-level biomass mapping. Carbon Balance and Management, 12

**Response:**

Thank you for your suggestion. We have revised it.

**L82:** Some recent studies integrated ground-based plot and remote sensing data for AGB mapping (Sun et al. 2011; Huang et al. 2019; Qi et al. 2019)

Sun, G., Ranson, K.J., Guo, Z., Zhang, Z., Montesano, P., & Kimes, D. (2011). Forest biomass mapping from lidar and radar synergies. Remote Sensing of Environment, 115, 2906-2916

Huang, W., Dolan, K., Swatantran, A., Johnson, K., Tang, H., O'Neil-Dunne, J., Dubayah, R., & Hurtt, G. (2019). High-resolution mapping of aboveground biomass for forest carbon monitoring system in the Tri-State region of Maryland, Pennsylvania and Delaware, USA. Environmental Research Letters, 14, 095002

Qi, W., Saarela, S., Armston, J., Ståhl, G., & Dubayah, R. (2019). Forest biomass estimation over three distinct forest types using TanDEM-X InSAR data and simulated GEDI lidar data. Remote Sensing of Environment, 232, 111283

**Response:**

Thank you for your suggestion. We have revised it.

**L85:** "First, existing studies that used machine learning methods have not considered the spatial heterogeneity of multiple environmental covariates (such as longitude, latitude, and forest structure)" This statement is too arbitrary. What does "structure" refer to? Shouldn't structure information came from lidar or radar?

**Response:**

Here, the forest structure refers to forest attributes such as stand volume, biomass, mean tree height, mean DBH, etc. It has been revised to "forest attributes like stand volume, biomass, mean height, et al."

**L96:** "multiple environmental covariates (such as longitude, latitude, and forest structure)" A duplicate statement, modify to be concise;

**Response:**

Thank you for your suggestion. We have deleted it.

**L140:** Suggest add equations of the allometric models you used here.

**Response:**

Thank you for your suggestion. We have added the model equation in the manuscript. Additionally, in the supplementary file, we have also added the specific parameters of the three models we established (Table B.3) as follows:

"AGB=a[(DBH)$^2$H]$^b$ "

| year | a | b |
|---|---|---|
| 1 ~ 2 | 0.1538 | 0.6993 |
| 3 ~ 5 | 0.0377 | 0.9244 |
| 6 ~ 10 | 0.0689 | 0.8489 |

**L175:** What software/package did you applied to construct your model?

**Response:**

We used R3.5.3 (https://www.r-project.org) to construct our model.

**L179:** (reference series)? Figure 3. (b) SVM and (c) RF are for classification, not regression;

**Response:**

Reference series means the corresponding reference series in Figure 3. For better understanding, we have revised it to " 'reference AGB data of plots' in Figure 3".

Thank you for your suggestion. Figure 3 has been revised.

[Figure]

L445: "we" should be "We".

**Response:**

Thank you for your suggestion. We have revised it.

**REFERENCES**

Qiu, Q. Y., Yun, G. L.,Zuo, S. D., Yan, J., Hua, L. Z., Ren, Y., Tang, J. F., Li, Y. Y., Chen, Q.: Variations in the biomass of eucalyptus plantations at a regional scale in southern china. Journal of Forestry Research, 29, 1263–1276, https://doi.org/10.1007/s11676-017-0534-0, 2018.

Djomo, A. N., Picard, N., Fayolle, A., Henry, M., Ngomanda, A., Ploton, P., McLellan, J., Saborowski, J., Adamou, I., and Lejeune, P.: Tree allometry for estimation of carbon stocks in African tropical forests, Forestry: An International Journal of Forest Research, 89, 446-455, https://doi.org/10.1093/forestry/cpw025, 2016.

Fayolle, A., Doucet, J.-L., Gillet, J.-F., Bourland, N., and Lejeune, P.: Tree allometry in Central Africa: Testing the validity of pantropical multi-species allometric equations for estimating biomass and carbon stocks, Forest Ecology and Management, 305, 29-37, https://doi.org/10.1016/j.foreco.2013.05.036, 2013.

Chave, J., Réjou-Méchain, M., Búrquez, A., Chidumayo, E., Colgan, M. S., Delitti, W. B. C., Duque, A., Eid, T., Fearnside, P. M., Goodman, R. C., Henry, M., Martínez-Yrízar, A., Mugasha, W. A., Muller-Landau, H. C., Mencuccini, M., Nelson, B. W., Ngomanda, A., Nogueira, E. M., Ortiz-Malavassi, E., Pélissier, R., Ploton, P., Ryan, C. M., Saldarriaga, J. G., and Vieilledent, G.: Improved allometric models to estimate the aboveground biomass of tropical trees, Global Change Biology, 20, 3177-3190, https://doi.org/10.1111/gcb.12629, 2014.

Picard, N., Rutishauser, E., Ploton, P., Ngomanda, A., and Henry, M.: Should tree biomass allometry be restricted to power models?, Forest Ecology and Management, 353, 156-163, https://doi.org/10.1016/j.foreco.2015.05.035, 2015.

---

## Author Comment (AC3) · 21 Apr 2020

The study presented in this manuscript compares the suitability of machine learning as well as geostatistical modelling approaches to retrieve maps of tree AGB on a regional scale based on plot-based surveys. The paper is reasonably written, however it is lacking a lot of important details that will be outlined in the more specific comments below.

Response:

Thank you for your affirmation and criticism of our manuscript. You have some misunderstanding about the manuscript, which shows that our article is not well written. The goal of the study was not to predict the forest AGB on a regional scale, but rather to improve the accuracy of the plot-scale AGB prediction model. We provide technical support for the mapping accuracy at the regional scale in the future. Plot scale is a bridge connecting single trees and regional scale. The rest of the specific questions are answered below.

**GENERAL COMMENTS**

1) Important information on the methodology are not given. In the method section it's not even mentioned which predictors were used for the machine learning model training. But this essential as the success of the models is only marginally depending on the choice of the algorithm but in the first place on the ability of the variables being used to serve as predictors for AGB! Only in the result section we get an idea on the variables (longitude, DBH, H, and forest age). I'm surprised about these variables as remote sensing information (especially NDVI) would present much more obvious predictors for AGB when the aim is to model AGB on a regional scale. With the selected variables, how could you upscale the results to a regional scale? I guess neither DBH nor H are not available in a spatial continuous way. So your model cannot be used for regional mapping! However, your motivation is to use it for regional modelling so my question is a) can you really do it with your approach and b) if yes, why are you not doing so and also show the results in the manuscript?

Response:

1. Answers to question regarding important information on the methodology.

We totally agree that the success of the model depends not only on the choice of the algorithm but also on the predictors of the model. Therefore, in the Materials and Methods section (2.4, Construction of the plot-level models), there are five subsections. In the first subsection (2.4.1, Selection of variables and analysis of resulting spatial distribution), the selection method for the prediction variables for the model is introduced in detail. Based on the previous results of our research group (doi:10.1007/s11676-016-0237-y), a series of environmental variables (including soil and topography variables) and forest attribute variables were selected, and Pearson's correlation analysis was used to test the correlation between them and our research objective (forest AGB) in order to select significant correlation variables as the prediction variables of the model. We have included the results of the variable selection in the Results section (3.2, Spatial distribution test and the selection of variables).

2. Answers to other questions related to the misunderstanding of the research purpose.

We are sorry that our manuscript was not written clearly enough to allow you to understand the goal of the study. We will revise the manuscript accordingly.

The purpose of our study was to improve the accuracy of the plot-level prediction model. Sample plots not only provide validation data for large-scale prediction but also provide multi-source environmental variable information to calibrate models. The AGB estimation at the plot level is a bridge connecting the accurate AGB measurement of single trees to the estimation of regional-scale AGB. Therefore, accurate AGB estimation at the plot level provides a basis for up-scaling to the regional level in the future. However, the uncertainty and error propagation inherent in different plot-level models make reliable up-scaling challenging. At present, allometric models are the most commonly used method to build AGB models at the plot level, however they cannot fully capture the impact of the complex spatial heterogeneity of multiple environmental covariates on the spatial distribution of AGB, nor can they adapt to the spatial dependence of model residual or the instability of model variables. The purpose of this study was to develop and evaluate a method that is superior to allometric models, that is, one involving a combination of machine learning and spatial statistics, in order to improve the accuracy of plot-level AGB prediction models.

Remote sensing information such as NDVI and other indicators are used in the next step of out of this manuscript, namely, the application of the optimal plot-level model. That includes (1) the

results of the optimal plot-level model provide more plot data (compared to only observed plot data) and more accurate plot data (compared to data from the allometric model) as input data and validation data for the prediction of a regional AGB map based on the wall-to-wall remote sensing data. This was performed to improve the accuracy of the plot-level model, which is the first step of your concern. Additionally, remote sensing data are not the only way to predict the AGB map over a large region. Forest inventory data at the national scale can provide the tree height, DBH, longitude. and other information needed for our model.

2) The cross-validation strategy that you used is not suitable if you have spatially clustered data (as you obviously have looking at the map). This is shown by several studies (see references below, to mention just a few). What would be appropriate is a spatial cross-validation that is testing the ability of your model to make predictions for spatially new samples. At least you should take care that you never use data points from the same forest patch for both training and testing. Otherwise, it is not possible to evaluate the ability of your models for regional mapping. Including coordinates as predictors when the data are spatially clustered is very dangerous (see Meyer et al 2019) and can lead to high overfitting which can only be revealed with spatial cross-validation. So I recommend that in addition to spatial cross-validation, to perform a spatial variable selection (i.e. can the predictors be used to make predictions for new locations?)

1. Meyer, H., Reudenbach, C., Wöllauer, S., Nauss, T., 2019. Importance of spatial predictor variable selection in machine learning applications - Moving from data reproduction to spatial prediction. Ecological Modelling. 411, 108815.

2. Pohjankukka, J., Pahikkala, T., Nevalainen, P., Heikkonen, J., 2017. Estimating the prediction performance of spatial models via spatial k-fold cross validation. Int. J. Geogr. Inform. Sci. 31, 2001–2019. https://doi.org/10.1080/13658816.2017. 1346255.

3. Roberts, D.R., Bahn, V., Ciuti, S., Boyce, M.S., Elith, J., Guillera-Arroita, G., Hauenstein, S., Lahoz-Monfort, J.J., Schröder, B., Thuiller, W., Warton, D.I., Wintle, B.A., Hartig, F., Dormann, C.F., 2017. Cross-validation strategies for data with temporal, spatial, hierarchical, or phylogenetic structure. Ecography.

4. Valavi, R., Elith, J., Lahoz-Monfort, J.J., Guillera-Arroita, G., 2018. blockcv: an r package for

generating spatially or environmentally separated folds for k-fold cross validation of species distribution models. BioRxiv.

Response:

1. Thank you very much for your suggestions. After studying the four references you recommended and considering carefully, we tested a spatial cross-validation method (blockCV) to evaluate our model's ability with and without the "longitude" variable. The new evaluation results were compared with the old ones, as shown in figures 1–3 below.

From the perspective of machine learning, each performance indicator value of blockCV is higher than that of the leave-one-out cross-validation (see figures 1–3 below: Figure 1 compared to Figure 2 and Figure 3 compared to Figure 2), which confirms your point of view. The cross-validation strategy we previously used may not be suitable as we have spatially clustered data. However, when we combined machine learning with spatial statistics, we found that there was no significant difference between the results of the two cross-validation strategies (see figures 1–3). This hints that the combination of spatial statistics and machine learning may alleviate the possible overfitting phenomenon to some extent. However, in our manuscript, we chose blockCV instead of leave-one-out cross-validation. The results and discussion will be revised one-by-one.

2. Furthermore, we investigated 22 additional plots, using the same predictors to test the seven models (see Section 2.4.5). We found that the observed results of model performance of these 22 plots was still stable (see Section 3.3 and Figure 7) when compares to the results of previous 30 plots. This proved the feasibility of the model to some extent.

Figure 1. blockCV (exclude longitude)

| | model | MAE | MRE | MAEsd | MREsd | RMSE | nRMSE | method | Type |
|---|---|---|---|---|---|---|---|---|---|
| 1 | SVM | 18.259926 | 2.2600814 | 18.724252 | 6.38939812 | 25.929442 | 0.5477280 | S1 | ML |
| 2 | ANN-RBF | 14.224423 | 0.3230649 | 15.124339 | 0.26072165 | 20.578022 | 0.4346857 | S2 | ML |
| 3 | RF | 16.401448 | 1.2910516 | 22.223087 | 3.23974756 | 27.320522 | 0.5771128 | S3 | ML |
| 4 | PBSHADE | 12.069387 | 0.2509008 | 15.564449 | 0.22142772 | 19.489667 | 0.4116955 | S4 | Sp Stats |
| 5 | SVM & PBSHADE | 5.074986 | 0.1084286 | 4.122087 | 0.04103023 | 6.494666 | 0.1371919 | S5 | ML&Sp Stats |
| 6 | ANN-RBF & PBSHADE | 11.601607 | 0.2548966 | 13.447962 | 0.18986444 | 17.590245 | 0.3715726 | S6 | ML&Sp Stats |
| 7 | RF & PBSHADE | 6.154559 | 0.1521969 | 10.085011 | 0.18624791 | 11.670295 | 0.2465208 | S7 | ML&Sp Stats |
| 8 | Allometric Model | 16.412314 | 0.5766961 | 19.288690 | 0.61802362 | 25.080188 | 0.5297885 | S8 | Allome |

Figure 2. Leave-one-outCV (include longitude)

| | Method | MAE | MRE | RMSE | maesd | mresd | | Type | nRMSE |
|---|---|---|---|---|---|---|---|---|---|
| 1 | SVM | 11.167837 | 0.2478704 | 10.387622 | 18.26950 | 0.26814000 | | ML | 0.2181603 |
| 2 | ANN-RBF | 12.148633 | 0.2669469 | 10.387868 | 18.28645 | 0.21892925 | | ML | 0.2181654 |
| 3 | RF | 10.155275 | 0.2593000 | 9.429181 | 16.46633 | 0.25093935 | | ML | 0.1980312 |
| 4 | P-BSHADE | 18.371450 | 0.3912975 | 14.077459 | 17.31066 | 0.16984299 | Sp | Stats | 0.2956540 |
| 5 | SVM&P-BSHADE | 6.882970 | 0.1246758 | 6.303799 | 11.00541 | 0.06971198 | ML&Sp | Stats | 0.1323920 |
| 6 | ANN-RBF&P-BSHADE | 10.135638 | 0.2049183 | 9.633279 | 16.89584 | 0.14301587 | ML&Sp | Stats | 0.2023176 |
| 7 | RF&P-BSHADE | 5.678865 | 0.1296616 | 5.299004 | 9.23540 | 0.12137309 | ML&Sp | Stats | 0.1112894 |

Figure 3. blockCV (include longitude)

```
> modeleval
          model       MAE        MRE     MAEsd        MREsd      RMSE     nRMSE method          Type
1           SVM   18.259926  2.2600814  18.724252  6.38939812  25.929442  0.5477280    S1          ML
2       ANN-RBF   14.664950  0.3244150  15.892523  0.23924533  21.429278  0.4526675    S2          ML
3            RF   16.434818  1.2918308  22.322241  3.25310871  27.418539  0.5791833    S3          ML
4       PBSHADE   12.069387  0.2509008  15.564449  0.22142772  19.489667  0.4116955    S4    Sp Stats
5  SVM & PBSHADE   5.074986  0.1084286   4.122087  0.04103023   6.494666  0.1371919    S5 ML&Sp Stats
6 ANN-RBF & PBSHADE 12.163764 0.2438393 12.861101 0.17556948  17.545697  0.3706315    S6 ML&Sp Stats
7    RF & PBSHADE   6.344222  0.1530373  10.692854  0.19070479  12.279048  0.2593800    S7 ML&Sp Stats
8 Allometric Model 16.412314 0.5766961 19.288690 0.61802362  25.080188  0.5297885    S8       Allome
```

**3)** The concerns outlined above can be improved, however, I have also doubts about the general value of the paper. Relying on 30 plots only is very very limited for machine learning application (the spatial CV will probably reveal this). So I doubt that the results will produce results that allow for general conclusions on the value of combining machine learning with geostatistical modelling.

**Response:**

We concede that our reports have to be interpreted cautiously, as they are limited to 30 plots. In our opinion, it is very valuable to investigate 30 plots with 90 constructive trees with different tree ages from the full lifecycle of *Eucalyptus*. Additionally, we investigated another 22 plots, and the observed trend of comparison among the models is still stable when compares to the results of previous 30 plots. We welcome other researchers to repeat our analysis on new datasets using our models. Expanding our research to more diverse datasets (possibly remote sensing data such as Lidar data or possibly more environmental data such as terrain, soil, and climate data) and also to a larger sample size.

We agree that the articles published in *Biogeosciences* should be universal and representative, and the research results can help to solve the hot issues in ecology and geosciences. Therefore, we understand your concern about the influence of our manuscript. We are sorry that we did not clearly explain the innovation and influences of our manuscript. We believe that our contribution has the

following values:

1. The innovation of our manuscript is to integrate machine learning and a spatial statistical model. The integration of these two can help to complement each other's advantages and improve the accuracy of AGB estimation models. Machine learning has the advantage of being able to handle complex and potentially nonlinear relationships between forest AGB and other variables. However, the initial samples of machine learning are randomly selected, which may lead to differences in the results of each operation of the model. Additionally, specific machine learning algorithms have their own disadvantages, such as RF uses the average value of all regression trees in the calculation, which may result in the overestimation of the lower value and the underestimation of the higher value. As opposed to machine learning, the P-BSHADE model (a spatial statistical model) takes into account the spatial autocorrelation and spatial heterogeneity of forest AGB and of environmental covariates, and remedies the bias of the observed values of the sampling plots in theory, which corresponds more to actual situations. A combined model takes the result of machine learning as the reference data (input data) of P-BSHADE, so that the fitting process of the combined model accounts more for spatial relationships than is the case for the single machine learning model. In addition to the theoretical advantages of these methods, case studies presented in this study also demonstrate the empirical superiority of the combined model.

2. Allometric model is a simple, fast, and universal equation that has been used in many studies. However, selection error in plot-level allometric modeling still leads to over 40% uncertainty (Djomo et al., 2016; Fayolle et al., 2013; Chave et al., 2014), and simple or complex forms of the allometric model account for 20% – 60% of the uncertainty (Picard et al., 2015). In our manuscript, we propose an improved method of AGB estimation which involves a combination of machine learning which is good at prediction and a spatial statistical model which is good at reflecting spatial relationships in order to improve the estimation accuracy of the AGB model at the plot level.

3. Over the past 20 years, with the growing area of *Eucalyptus* plantations around the world, Brazil, India, China, Chile, Spain, DR Congo, Australia, South Africa, and other countries have established contiguous *Eucalyptus* planting areas. There have been many studies and reports on

the biomass estimation of *Eucalyptus* plantations using ecological process models at different temporal and spatial scales. China is the country with the largest area of planted forest in the world. However, China's planted forest suffers from the three practical problems of low productivity, unsustainability, and incongruous production function and ecological function, which urgently need to be solved by appropriate management measures. On the one hand, with the rapid growth of the global population, the timber demand is also increasing rapidly. On the other hand, there is the severe reality that total global forest resources have declined sharply in recent years. Thus, many countries and regions are vigorously developing fast-growing non-native trees to alleviate the contradiction between the supply and demand of timber and forest products in order to maintain economic and social development. *Eucalyptus* is one of the fastest-growing trees, and is controversial in the development of planted forest. A special study on Eucalyptus is of great significance.

4. Although *Eucalyptus* is taken as a case study in our manuscript, the model we proposed can also be applied to other tree species or mixtures of tree species. There is no particular relationship between our model settings and tree species. No unique characteristics of *Eucalyptus* is added to the model, and other forest types also can provide the input data such as tree height, DBH, longitude, and other variables. Therefore, the model we proposed can be expanded and its influence is not limited to *Eucalyptus* forests. In fact, this idea of improving the accuracy of the model also has the potential to solve other ecological problems.

In view of the influence of the four aspects of the manuscript we described above, we believe it can attract attention

**SPECIFIC COMMENTS**

**Line 25:** I disagree that longitutde and latitude on this scale affect AGB. Even on a large scale they don't but are just proxies for e.g. climate but they are certainly not underlying factors for AGB on your small study area.

**Response:**

Perhaps you have mainly considered the direct influence of direct factors. However, in ecology,

the influence of indirect factors is also very important and should not be ignored. Forest ecosystems are complex systems. The relationship between forest biomass and the surrounding environment also deserves attention on a small scale. How forest biological factors and abiotic environmental factors affect the distribution and estimation of forest biomass has not been studied thoroughly. In the field of forest biomass estimation, many other scholars have emphasized and attached importance to the potential of topographic factors for improving model estimation (e.g., Fassnacht et al., 2014). It is necessary to consider the interaction and the nonlinear and indirect effects of these environmental factors to improve the accuracy of forest biomass estimation. We referred in particular to the fact that latitude and longitude are not suitable in here, and therefore we changed them for topographic, soil, and climatic information. Additionally, as the comparison with or without the longitude factor as the predictor yielded similar model performance (see Figures 1 and Figures 3 above: Figure 1 compared to Figure 3), we removed the longitude from the predictors.

**Line 49-51:** One important thing is missing: The model might also fail because the predictor variables are not sufficient to estimate AGB.

**Response:**

Thank you for your reminder. We have included it as one of the sources of model-dependent uncertainty. The modifications are as follows:

*"The uncertainty of such regional maps can be attributed to three primary sources: (1) the use of inadequate sampling data to construct the plot level prediction models, (2) model-dependent uncertainty, including unreasonable model-parameter assumptions, improper model structure (Chen et al., 2015; Gao et al., 2016; McRoberts et al., 2016),* **and the predictor variables are not sufficient to estimate AGB (Meyer et al., 2019).** *The present study mainly focuses on reducing the second source of uncertainty."*

**Line 54:** "An estimated 18%–103% of the uncertainty in AGB mapping can be attributed to model-dependent uncertainty". In fact between nearly nothing (~18) and everything (>100). That sounds unreasonable, consider taking that sentence out.

**Response:**

Thank you for your suggestion. We have modified the expression as follows:

"Up to 103% of the uncertainty in AGB mapping can be attributed to model-dependent uncertainty".

**Line 60-62:** This differentiation between allometric models and statistical models does not seem to make sense. E.g. Allometric models can be based on linear relationships as well. Please improve the logical structure here.

**Response:**

We were referring to the spatial statistical model, generally the geostatistical techniques, such as geographically weighted regression (GWR), ordinary least-squares regression (OLS), and so on. They are different from allometric models. However, there are logical problems. As you said, allometric models can be based on linear relationships as well. Therefore, we revised the expression as follows:

*"Many different plot-level prediction models other than allometric models have been applied to constructing accurate AGB maps, including **other** linear models (Andersen et al., 2014; Morel et al., 2012), machine learning models (Chen, 2015; Gleason and Im, 2012), and spatial statistical models (Benitez et al., 2016; Propastin, 2012;Van der Laan et al., 2014)."*

**Line 67-71:** Be careful with the logical structure here as well: The major advantage is that machine learning is able to fit complex relationships which e.g. linear models don't. And THEREFORE they might be advantageous in predictions (not "in addition" as you write in Line 72).

**Response:**

Thank you for your suggestion. We have modified the expression as follows:

*"By comparison, nonparametric machine learning algorithms, in which the number of parameters depends on the number of training examples (e.g., K-nearest neighbor, support vector machine, and random forest), are advantageous because they are more elastic and do not restrict variable types, the distribution of predictor variables, or the relationship between response and predictor variables (Lu et al., 2007). **Therefore,** nonparametric machine learning algorithms may offer higher prediction accuracy (Frey et al., 2019; Gleason and Im, 2012)."*

**Line 74-81**: The fundamental difference between the approaches is not getting clear here but this is important because combining the two approaches is the objective of the paper. In contrast to the statistical (including machine learning) approaches explained above, the spatial statistical approaches have the major assumption that "near things are more related that distant things". I think the general idea should be made clear and it should be explained why you expect that a combination might be the way forward.

**Response:**

Your question does not correspond to the number of lines shown. In lines 74–81, we briefly introduced some research and applications of spatial statistical models to the study of the relationship between forest AGB and multi-source environmental factors and compared the advantages of the spatial statistical model and the traditional statistical model. The two methods which were combined in our study were machine learning and spatial statistics. Lines 99–104 show why we hope the combined method might be the way forward.

Lines 74-81 read as follows: *"Another group of models frequently used to estimate the relationship between forest AGB and multiple environmental covariates is based on spatial statistical approaches, including geographically weighted regression and Kriging (Du et al., 2010; Van der Laan et al., 2014; Viana et al., 2012). Spatial statistical methods are based on analyses of attribute information, such as spatial location (Schabenberger and Gotway, 2005). Compared with traditional statistical methods, spatial methods integrate spatial factors that affect model responses, thus removing the constraints of traditional statistical methods that assume sample independence (Rangel and Bini, 2010) and improving our understanding of spatial autocorrelation and heterogeneity (He et al., 2011; Rosenberg and Anderson, 2011)."*

Lines 99-104 read as follows: *"The proposed method integrates the nonlinear mapping capabilities of machine learning algorithms [i.e., radial basis function artificial neural network (RBF-ANN), support vector machine (SVM), and random forest (RF)] with the spatial autocorrelation and stratified heterogeneous advantages of a spatial statistical model (i.e., the point estimation model of biased sentinel hospital-based area disease estimation, P-BSHADE) (Xu et al., 2013)."*

**Line 85-88** "studies that used machine learning methods have not considered the spatial

heterogeneity of multiple environmental covariates (such as longitude, latitude, and forest structure)".
I disagree. Most approaches use environmental covariates which of course have been heterogeneous as well. No information on model tuning is given. Also please state which software implementations and settings of the algorithms you used.

Response:

1. Thank you for your suggestion. We have revised the inappropriate expression as follows:

*"**some existing** studies that used machine learning methods have not considered the spatial heterogeneity of multiple environmental covariates (such as longitude, latitude, and forest structure)".*

2. The information on model tuning includes: (1) SVM: (type="eps-regression",kernel="radial",cost=10,gamma=0.2); (2) RF: (size=3,maxit=2500). The rest are default parameter settings. We will list the model code in the supplementary material.

3. Using R 3.5.3 (https://www.r-project.org) to implement the algorithms.

**Line 157-158:** please explain why you tested for spatial autocorrelation etc. Why is this information relevant for the modelling?

Response:

1. Testing spatial autocorrelation and spatial heterogeneity are the premises of applying the P-BSHADE spatial statistical model. It assumes that the research object has spatial autocorrelation and spatial heterogeneity, so we must detect the spatial autocorrelation and spatial heterogeneity of the research object (forest AGB in our manuscript) before using the P-BSHADE model.

2. P-BSHADE is a spatial statistical inference model. P-BSHADE is an optimal linear unbiased estimation interpolation method based on the assumption of the simultaneous existence of the spatial autocorrelation and heterogeneity of the target object.

**Line 205:**"Because of the Law of Large Numbers, RF does not overfit." That's wrong! Maybe random forest is robust to overfitting in terms of hyperparameter selection but it is not the case if you have data that are not independent. See e.g. the references mentioned above.

Response:

Thank you for your professional guidance in machine learning. The incorrect description has been deleted.

**Line 206-207:** Accurate predictions of random forest do NOT in the first place originate from injecting randomness. E.g. If the predictors are not sufficient to estimate a response variable, random forest will fail (and so will other algorithms)!

Fig. 6 : Is this based on the cross-validation?

Response:

1. Thank you for your professional guidance in machine learning. The incorrect description has been deleted. We agree with you that predictors are important to the model. This manuscript focused on the comparison and screening of different models. The same predictors are used in all the machine learning models used in our study. The determination of predictors was based on the results of Pearson correlation analysis. However, since this manuscript did not compare different predictors of the accuracy of the model, we may consider this in future work.

2. Yes, based on the cross-validation.

**Line 502-503:** "The assumption is that estimated AGB is accurate in all sampling plots except the target sampling plot. In other words, the premise behind using only the P-BSHADE model is that the reference AGB data is accurate or strongly correlated with AGB. " I don't understand that. The same reference data were used for all modeling approaches and for sure we assume that the reference data are accurate for both types of models.

Response:

P-BSHADE needs the reference value of the target object (which is forest AGB in our manuscript) as input data. Although we assumed that the reference value is accurate, there are many ways to obtain the reference value. On one hand, how to obtain the reference value is the key to combining other methods with the P-BSHADE model, and therefore this leads to differences among the different combined models in our manuscript. On the other hand, we used an allometric model to obtain the reference data of the single P-BSHADE, which leads to differences between the single P-BSHADE model and combined models. The specific steps included: (1) we first used the machine

learning method or allometric method to estimate AGB and then (2) used the estimated AGB as a reference value in the P-BSHADE model to estimate AGB again.

Therefore, when comparing combination models with the single P-BSHADE model in the discussion, it is logical to compare and illustrate their reference values.

**Line 578:** "We used FMPI data to upscale the optimal plot-level AGB model from plot level to region scale." Did you? We don't get to see the results for the regional upscaling. I wonder: Is your model really better than simply using the average measured AGB from each forest site as estimate for AGB for the entire patch?

**Response to the first question:** Yes, we did this work, but as it is not the main objective of our manuscript, we put this result in Section 6 of the supplementary material (Figures C.3 and C.4).

**Response to the second question:** We do not quite understand the method you want to compare with our optimal model. Do you mean simply using the average value of AGB measured at all forest sites as the estimated AGB of the whole forest class? This method may be relatively simple, and may have advantages in specific circumstances, such as when the accuracy requirements are not high. However, we insist that our approach is better for the following reasons:

1. The purpose of our study is to improve the accuracy of the plot-level AGB estimation model. This goal is an important prerequisite to improving the accuracy of regional AGB estimation because regional AGB estimation often needs to use the verification data and training data provided by sample plots.

2. The way to achieve this goal is to combine machine learning with spatial statistics and make use of their complementary advantages to establish an optimal model with high accuracy for the estimation of AGB in sample plots.

**REFERENCES**

Chave, J., Réjou-Méchain, M., Búrquez, A., Chidumayo, E., Colgan, M. S., Delitti, W. B. C., Duque, A., Eid, T., Fearnside, P. M., Goodman, R. C., Henry, M., Martínez-Yrízar, A., Mugasha, W.

A., Muller-Landau, H. C., Mencuccini, M., Nelson, B. W., Ngomanda, A., Nogueira, E. M., Ortiz-Malavassi, E., Pélissier, R., Ploton, P., Ryan, C. M., Saldarriaga, J. G., and Vieilledent, G.: Improved allometric models to estimate the aboveground biomass of tropical trees, Global Change Biology, 20, 3177-3190, https://doi.org/10.1111/gcb.12629, 2014.

Djomo, A. N., Picard, N., Fayolle, A., Henry, M., Ngomanda, A., Ploton, P., McLellan, J., Saborowski, J., Adamou, I., and Lejeune, P.: Tree allometry for estimation of carbon stocks in African tropical forests, Forestry: An International Journal of Forest Research, 89, 446-455, https://doi.org/10.1093/forestry/cpw025, 2016.

Fassnacht, F. E., Hartig, F., Latifi, H., Berger, C., Hernández, J., Corvalán, P., and Koch, B.: Importance of sample size, data type and prediction method for remote sensing-based estimations of aboveground forest biomass, Remote Sensing of Environment, 154, 102-114, 2014.

Fayolle, A., Doucet, J.-L., Gillet, J.-F., Bourland, N., and Lejeune, P.: Tree allometry in Central Africa: Testing the validity of pantropical multi-species allometric equations for estimating biomass and carbon stocks, Forest Ecology and Management, 305, 29-37, https://doi.org/10.1016/j.foreco.2013.05.036, 2013.

Picard, N., Rutishauser, E., Ploton, P., Ngomanda, A., and Henry, M.: Should tree biomass allometry be restricted to power models?, Forest Ecology and Management, 353, 156-163, https://doi.org/10.1016/j.foreco.2015.05.035, 2015.

---

## Referee Comment (RC2) · Anonymous Referee #2 · 16 May 2020

Dear editors, dear authors,

The manuscript 'improving maps of forest aboveground biomass: A combined approach using machine learning with a spatial statistical model' by Dai et al. presents a new approach to predict more accurately the Aboveground biomass.

They do so by combining a statistical approach, the P-BSHADE model, with machine learning models. They claim that the joint model approach is superior to machine learning models and the P-BSHADE model alone.

I found the general ideas to use machine learning for such a predictive task and to combine machine learning models with statistical models very appealing and I think that the

community would benefit strongly from an approach capable of predicting accurate the AGB at scale.

General comments:

As I am not part of the remote sensing community I cannot say much about the novelty of this approach, but I have a good experience with machine learning methods and I will focus on the methodological part of this manuscript.

a) I must say that the paper is sloppy in multiple respects, many statements about machine learning are inaccurate and partly wrong. Especially in the third paragraph of their introduction (L60-L73), many of their claims about machine learning are only partially true or are confusing (see specific comments): they say 'nonparametric machine learning algorithms, in which the number of parameters depends on the number of training examples', however, if they are nonparametric how can their number of parameters depend on anything? Also, the authors use RF, ANN, and SVM in their work as regression models but why do they explain and illustrate them as classifier? (see method section and Fig. 3). In summary, the authors should carefully revise all their statements about ML and explain correctly their used ML models.

b) They claim that the joint model combines the advantages of ML and the P-BSHADE model, the predictive non-linearity advantage of ML and the ability of the P-BSHADE to capture spatial relationships. However, if they are given the chance I think that ML models are also capable of detecting and using spatial relationships, that means, you have to provide them not only longitude but also latitude as predictor! Based on correlation with AGB, the authors selected only longitude, however, I would assume that an interaction of longitude and latitude would be a good predictor of spatial relationships (two variables of an interaction can show by themselves low correlation). Moreover, ML models such as RF are outstanding in detecting interactions and higher-order interactions (if they are given the chance). Also, hyper-parameter tuning is important in ML to improve predictive performance, even for RF! (e.g. see Probest et al., 2019

https://doi.org/10.1002/widm.1301). I recommend that the authors re-evaluate the performance of the ML models with hyper-parameter tuning, nested cross-validation, and additional predictors (at least latitude).

c) I found it very distressing that section about the P-BSHADE model (L241-280) was taken almost literally from a previous work by one of the co-authors (Xu et al., 2013)! An illustration: Xu et al. 2013 (https://doi.org/10.1175/JCLI-D-12-00633.1): 'This equation is generally valid for a nonhomogeneous condition. Clearly, determination of $\hat{y}0$ requires calculation of coefficients $w_{ij}$ (. . .), which is addressed in the following section'. . . in the MS: 'This equation is generally valid for nonhomogeneuous conditions. Clearly, the determination of $b_{ij}$ requires calculating the coefficients $w_{ij}$ (. . .), which is addressed in the following section.'

Specific:

L 28: I suggest that you re-position the following sentence in abstract. 'The study was conducted' should come after the introduction of our methods

L39: Sentence is redundant

L52: 'the present study. . .'

L63-L65: not the development of computer-science techniques but the advances in hardware are responsible for the popularity of ML. Most of the ML techniques are quite old (e.g. Artificial neural networks, even CNNs).

L65: "which summarize data with a fixed number of parameters based on sample size". This statement is inaccurate or even wrong (it is difficult to understand the authors' intention). A) "summarize data" is wrong, or what do you mean? B) fixed number of parameters based on sample size; I think this is wrong because fitting 'parametric' models with p » n is a common task/problem with well-known solutions (e.g. regularization/elastic net). Moreover deep neural networks are highly parametrized models and are not 'non-parametric'. Here, you should focus on the distinction between linear

and non-linear models.

L68: remove 'nonparametric' and use non-linear

L68-69: What do you mean with that the number of parameters on the number of training examples? This statement conflicts with your previous statement L65 and why or how does the number of parameters depend on n in kNN, SVM, and RF?

L70: restrict variable types? A linear regression is not restricted to specific data types. Actually, kNN and SVM require the same contrasts as a linear regression and only RF is able to handle non-contrasted categorical predictors

L71: What do you mean with the distribution of predictor variables? I think that kNN and SVM are indeed affected by the distribution of the predictors because they use distance measurements.

L90: Why? Or at least provide a reference

L97: Please provide example references

L187: 'Each model was trained on 30 datasets. . .' But within the CV, right? So it should be 'trained on 29 datasets'.

L199-200: Not exactly true, the activation function makes the transformation linear or non-linear. The fundamental matrix multiplication is a linear function.

L197-199: Is there a reason you use an RFB ANN? You could also use a normal DNN with relu activation functions and several hidden layers. Could you also explain the RBF-ANN in detail? I think that most users do not know how the RBF function is used by RBF-ANN.

L205: Wrong, RF can overfit, I thin with the law of large numbers you refer to the number of trees which is true that increasing the number of tree does not increase the generalization error. But if we assume that we have a sufficient number of trees, RF can overfit.

L227: What are these "obvious advantages"?

L248: 'when j = 1, i = 2, 3, . . .30: when j = 1, i = 1, 3, 5, . . ., 30)' What do you mean?

L241-280: The description of the P-BSHADE model is too close to the original work Xu et al. 2013! Either you rewrite it completely in your own words, or which I suggest is that you try to summarize the method and move a detailed description to the Appendix. The P-BSHADE model is not the focus of your work.

L312: 'A detailed description of the combined models. . .' is missing in the Supplementary Material

L347: Because of a low correlation you did not choose latitude, however, I hypothesize that the interaction between longitude and latitude has an effect on AGB, which you would not see in the correlation table unless to test explicitly the correlation between the interaction and the AGB.

L479: 'Machine learning models appear adept at tackling high-dimensional problems.' Yes, they do, but you do not have a 'high-dimensional problem'.

L511: '. . . regression trees . . .' this applies only for RF and I wonder if the RF really suffer from a skewed response distribution. Do you have a reference?

L568: RF and SVM are also sensitive to hyper-parameters. It is myth that RF does not need hyper-parameter tuning (see https://doi.org/10.1002/widm.1301).

Figures:

Fig 3: The sematic figures of the ML do not fit to the way you used them in your work: RBF-ANN, RF, and SVM are illustrated as classifier with 4, 3, and 2 response classes, however, you use them as regression models (e.g. one output node for the RBF ANN, no majority voting for the RF etc.)

Fig 6: Revise your color choice, e.g. it is difficult to distinguish between the yellow and blue line.

[Figure]

---

## Author Comment (AC4) · 5 Jun 2020

Dear Editor,

Thank you for your email and the reviewer's comments concerning our manuscript entitled "Improving maps of forest aboveground biomass: A combined approach using machine learning with a spatial statistical model" (ID: bg-2020-36). We thank the reviewer for the very helpful comments.

**Detailed reply to reviewer's comments:**

**Comment 1:** The manuscript 'improving maps of forest aboveground biomass: A combined approach using machine learning with a spatial statistical model' by Dai et al. presents a new approach to predict more accurately the Aboveground biomass.

They do so by combining a statistical approach, the P-BSHADE model, with machine learning models. They claim that the joint model approach is superior to machine learning models and the P-BSHADE model alone.

I found the general ideas to use machine learning for such a predictive task and to combine machine learning models with statistical models very appealing and I think that the community would benefit strongly from an approach capable of predicting accurate the AGB at scale.

**Response to comment 1:** We very much appreciate the positive comments on our manuscript from Anonymous Reviewer #2. The reviewer's efforts have helped us greatly improve this paper.

**General comments:**

**Comment 2:** As I am not part of the remote sensing community I cannot say much about the novelty of this approach, but I have a good experience with machine learning methods and I will focus on the methodological part of this manuscript.

a) I must say that the paper is sloppy in multiple respects, many statements about machine learning are inaccurate and partly wrong. Especially in the third paragraph of their introduction (L60-L73), many of their claims about machine learning are only partially true or are confusing (see specific comments): they say 'nonparametric machine learning algorithms, in which the number of parameters depends on the number of training examples', however, if they are nonparametric how can their number of parameters depend on anything? Also, the authors use RF, ANN, and SVM in their work as regression models but why do they explain and illustrate them as classifier? (see method section and Fig. 3). In summary, the authors should carefully revise all their statements about ML and explain correctly their used ML models.

**Response to comment 2:** Thank you for your valuable comments, which have significantly improved our manuscript. Your opinions on the ML methods and their presentation in the paper are very useful and important. Thanks for pointing out the mistakes we made about machine

learning in the previous version. We have carefully and thoroughly reviewed and modified this article, especially the presentation of machine learning methods (see specific responses below).

The definitions of parametric and nonparametric models are sourced from Statistics and may be confused with some concepts in Computer Sciences. Here, we have carefully followed the definitions of 'parametric models' and 'nonparametric models' from a classical textbook on artificial intelligence titled 'Artificial Intelligence: A Modern Approach (3rd edition)'. Page 737 of this book reads, 'A learning model that summarizes data with a set of parameters of fixed size (independent of the number of training examples) is called a parametric model.' Page 737 reads, 'A nonparametric model is one that cannot be characterized by a bound set of parameters. For example, suppose that each hypothesis we generate simply retains within itself all of the training examples and uses all of them to predict the next example. Such a hypothesis family would be nonparametric because the effective number of parameters is unbounded — it grows with the number of examples. This approach is called instance-based learning or memory-based learning.' kNN, SVM, and RF all belong to this instance-based learning approach.' Page 758 reads, 'Nonparametric models use all the data to make each prediction, rather than trying to summarize the data first with a few parameters. Examples include nearest neighbors and locally weighted regression.'

Nonparametric ML methods are called 'nonparametric' not because there are no parameters in the model; indeed, nonparametric ML models typically have one or more hyperparameters (external parameters of the model whose values cannot be estimated from the data) plus a number of common parameters (internal parameters whose values can be estimated using the data). Examples of nonparametric ML methods include k-nearest neighbors, decision trees, and nonlinear support vector machines.

Whether ANN is a parametric or nonparametric method is debatable. As the number of ANN parameters grows with the number of neurons/layers it is therefore considered as nonparametric in some references (e.g., Orr 1996, which is a basic and classical paper for radial basis function networks and has been cited more than 1000 times). This work follows Orr (1996) and considers RBF-ANN as a nonparametric model. We agree with you that we also need to describe the advantages of these nonparametric models in dealing with nonlinear fitting, especially in our forest ecology nonlinear systems.

References

Orr, M. J. L.: Introduction to radial basis function networks. University of Edinburgh, 1996.

Russell, S. J, & Norvig, P.: Artificial intelligence: a modern approach, 3rd edition, Pearson Publisher, 2016.

List of changes to machine learning:

1. The third paragraph of the Introduction; see the box immediately below 'List of changes to machine learning' and the response to comments 8–13.

2. The Methods section '(1) Machine learning' part of Section 2.4.3 ('Model training'); see the box immediately below 'List of changes to machine learning' and the response to comments 17–19.

3. We deleted the inappropriate statement about machine learning in Section 4.2 ('Model comparisons') of the Discussion, and modified references; see the box immediately below 'List of changes to machine learning' and the response to comments 25–27.

4. We have revised Figure 3; see the response to comment 28.

For your convenience, the revised text is reproduced below:
* * *
1. The third paragraph of the Introduction:

With advances in hardware and the further requirement of forest biomass modeling accuracy, machine learning methods have become prevalent in the field of forest biomass estimation. Traditional parametric methods, which summarize data with a fixed number of parameters regardless of sample size (e.g., logistic regression and perceptron) (Liu et al., 2020; Russell & Russell, 2016), usually require prior knowledge on a system to be modelled and assume a specific function for the system. Then the training data is used to estimate parameters of the presumed function. By comparison, nonparametric machine learning algorithms, which do not need sufficient understanding on the systems to be modelled, but seek the best fitting of the training data. They cannot be characterized by a fixed number of parameters (the number of parameters depends on the number of training examples, e.g., k-nearest neighbor, nonlinear support vector machine, and random forest). As a result, they are more elastic, can fit different function forms without prior knowledge, and achieve excellent fit performance (Russell and Norvig, 2016). Moreover, these nonlinear nonparametric models have advantages in dealing with nonlinear fitting and have great application potential in nonlinear systems such as forest ecology. Therefore, nonparametric machine learning algorithms may offer higher prediction

accuracy in forest AGB estimation (Frey et al., 2019; Gleason and Im, 2012).

Frey, U. J., Klein, M., and Deissenroth, M.: Modelling complex investment decisions in Germany for renewables with different machine learning algorithms, Environmental Modelling & Software, 118, 61-75, https://doi.org/10.1016/j.envsoft.2019.03.006, 2019.

Gleason, C. J., and Im, J.: Forest biomass estimation from airborne LiDAR data using machine learning approaches, Remote Sensing of Environment, 125, 80-91, 2012.

Liu, B., Gao, L., Li, B., Marcos-Martinez, R., and Bryan, B. A.: Nonparametric machine-learning as an effective tool for mapping forest cover and exploring influential factors, Landscape Ecology, https://doi.org/10.1007/s10980-020-01046-0, 2020.

Russell, S. J, & Norvig, P.: Artificial intelligence: a modern approach, 3[rd] edition, Pearson Publisher, 2016.

2. The Methods section '(1) Machine learning', part of Section 2.4.3 ('Model training'):

SVM is a method of supervised learning in machine learning which is often used to solve classification problems and can also be used to solve regression problems. The basic principle of SVM for classification is to find a hyperplane in the feature space and separate the positive and negative samples with the minimum misclassification rate (Hearst et al., 1998). The principle of SVM for regression is very similar to that of SVM for classification, with only slight differences (Sayad, 2020). The SVM for regression keeps the main idea of minimizing error, individualizing the hyperplane which maximizes the margin, and part of the error is acceptable (SaedSayad, 2020). RBF-ANN is a three-layer neural network model which includes an input layer, a hidden layer (a Gaussian RBF is used here), and an output layer. The transformation from input space to hidden space is linear or nonlinear, whereas the transformation from hidden space to output space is linear. The function of the hidden layer is to map the vector from the indivisible low-dimensional linear state to the separable high-dimensional linear state so as to greatly accelerate the learning and convergence speed and avoid getting stuck in a local optimum (Elanayar and Shin, 1994). RF is a relatively new machine learning technique and data mining method which combines self-learning technologies and was developed by Breiman in 2001. RF is a combination of tree predictors such that each tree depends on the values of a random vector that is sampled independently and with the same distribution for all trees in the forest. RF is effective in prediction (Breiman, 2001).

Breiman, L.: Random forests, Machine Learning, 45, 5-32, 2001.

Elanayar, V. T. S., and Shin, Y. C.: Radial basis function neural network for approximation and estimation of nonlinear stochastic dynamic systems, IEEE Transactions on Neural Networks, 5, 594-603, 1994.

Hearst, M. A., Dumais, S. T., Osman, E., Platt, J., & Scholkopf, B.: Support vector machines, IEEE Intelligent Systems, 13, 18-28, 1998.

Sayad S.: https://www.saedsayad.com/support_vector_machine_reg.htm, last access: 02 June 2020.

3. Modified reference:

Probst, P., Wright, M. N., Boulesteix, A-L.: Hyperparameters and tuning strategies for random forest. WIREs Data Mining Knowledge Discovery. e1301, https://doi.org/10.1002/widm.1301, 2019.

4. Revised Figure 3:

[Figure]

**Comment 3:**  b) They claim that the joint model combines the advantages of ML and the P-BSHADE model, the predictive non-linearity advantage of ML and the ability of the P-BSHADE to capture spatial relationships. However, if they are given the chance I think that ML models are also capable of detecting and using spatial relationships, that means, you have to provide them not only longitude but also latitude as predictor! Based on correlation with AGB, the authors selected only longitude, however, I would assume that an interaction of longitude and latitude would be a good predictor of spatial relationships (two variables of an interaction can show by themselves low correlation). Moreover, ML models such as RF are outstanding in detecting interactions and higher-order interactions (if they are given the chance). Also, hyper-parameter tuning is important in ML to improve predictive performance, even for RF! (e.g. see Probest et al., 2019 https://doi.org/10.1002/widm.1301). I recommend that the authors re-evaluate the performance of the ML models with hyper-parameter tuning, nested cross-validation, and additional predictors (at least latitude).

**Response to comment 3:** Many thanks for your valuable recommendations to improve this work. We agree that ML models can also detect and use spatial relationships. However, it is not the focus of this work to improve the performance through optimizing the structures of ML models or tuning hyper-parameters. As forest ecology modelers, we are more interested in the performance of combining machine learning with a spatial statistical method in forest AGB estimation as previous literature has shown that such a combined method can further improve the predictive performance in the spatial interpolation of environmental variables (Li et al., 2011). Therefore, the scientific question this work attempts to answer is: Given the limited expertise/knowledge of non-ML-community practitioners to further improve ML performance, can we further improve the predictive performance by combining machine learning with a spatial statistical method?

In the previous version, we did not include latitude as a predictor because it is controversial whether longitude and latitude should be used as predictors, and some research efforts found that including them could lead to considerable overfitting (e.g., Meyer et al., 2019). Moreover, as suggested, we have tested the performance of models by including both longitude and latitude as predictors, and compared the models with those that only include longitude or neither longitude nor latitude (see **response to comment 24** for detailed results). The results showed that including longitude and latitude as predictors did not improve the performance of RF, SVM, or RBF-ANN.

However, we acknowledge that your recommendations are valuable. We have discussed your recommendations in the revised manuscript, including them as future directions/limitations of this work. We have also read and cited the reference you kindly provided.

For your convenience, the revised text is reproduced below:

> In this paper, we use a correlation analysis to determine which predictors are included to train a model, without considering whether the interaction among predictors has an impact on the forest AGB. Although we found that the interaction of longitude and latitude did have an impact on the AGB, our experiments also found that including them as predictors simultaneously did not improve the performance of the machine learning models. This may be due to two reasons: 1) we have not found the specific interaction relationship between longitude and latitude. In the future, we will test whether the different interaction patterns between these two predictors can improve the performance of machine learning in forest AGB estimation; and 2) in this work, we manually adjusted the hyper-parameters and did not fully explore the potential of machine learning methods. In future work, we will improve the performance of model

estimation through optimizing the structures of machine learning models or tuning hyper-parameters (Probest et al., 2019).

Li, J., Heap, A. D., Potter, A., and Daniell, J. J.: Application of machine learning methods to spatial interpolation of environmental variables, Environmental Modelling & Software, 26, 1647-1659, https://doi.org/10.1016/j.envsoft.2011.07.004, 2011.

Meyer, H., Reudenbach, C., Wöllauer, S., Nauss, T.: Importance of spatial predictor variable selection in machine learning applications - Moving from data reproduction to spatial prediction. Ecological Modelling, 411, https://doi.org/10.1016/j.ecolmodel.2019.108815, 2019.

Probst, P., Wright, M. N., Boulesteix, A-L.: Hyperparameters and tuning strategies for random forest. WIREs Data Mining Knowledge Discovery. e1301, https://doi.org/10.1002/widm.1301, 2019.

**Comment 4:** c) I found it very distressing that section about the P-BSHADE model (L241-280) was taken almost literally from a previous work by one of the co-authors (Xu et al., 2013)! An illustration: Xu et al. 2013 (https://doi.org/10.1175/JCLI-D-12-00633.1): 'This equation is generally valid for a nonhomogeneous condition. Clearly, determination of $y^{\check{}}0$ requires calculation of coefficients wij (. . .), which is addressed in the following section'. . . in the MS: 'This equation is generally valid for nonhomogeneuous conditions. Clearly, the determination of bij requires calculating the coefficients wij (. . .), which is addressed in the following section.'

**Response to comment 4:** We are deeply sorry for that. Lines 241–280 described the specific equations for the P-BSHADE. In order to make this mew method understandable in our forest AGB estimate, we only changed the descriptions of specific symbols in the equations but did not modify other important descriptions, such as the sentences you listed above, in order to avoid misunderstanding. We apologize for this and accept your kind advice in comment 22. Therefore, we have summarized the P-BSHADE method in the manuscript and have given detailed information in the Supplementary Material.

**Specific comments:**

**Comment 5:** L28: I suggest that you re-position the following sentence in abstract. *'The study was conducted'* should come after the introduction of our methods

**Response to comment 5:** Thank you for your professional advice. We have put this sentence after the introduction of the methods.

**Comment 6:** L39: Sentence is redundant

**Response to comment 6:** Thank you for your kind advice. We have deleted the sentence.

**Comment 7:** L52: *'the present study. . .'*

**Response to comment 7:** Thank you for your kind advice.

**Comment 8:** L63-L65: not the development of computer-science techniques but the advances in hardware are responsible for the popularity of ML. Most of the ML techniques are quite old (e.g. Artificial neural networks, even CNNs).

**Response to comment 8:** Thanks for your kind advice. We have revised this sentence as follows:

With advances in hardware and the further requirement of forest biomass modeling accuracy, machine learning methods have become prevalent in the field of forest biomass estimation.

**Comment 9:** L65: *"which summarize data with a fixed number of parameters based on sample size"*. This statement is inaccurate or even wrong (it is difficult to understand the authors' intention). A) *"summarize data"* is wrong, or what do you mean? B) fixed number of parameters based on sample size; I think this is wrong because fitting *'parametric'* models with p *»* n is a common task/problem with well-known solutions (e.g. regularization/elastic net). Moreover deep neural networks are highly parametrized models and are not *'non-parametric'*. Here, you should focus on the distinction between linear and non-linear models.

**Response to comment 9:** Thanks for this comment.

Here, we adopted the definition of 'parametric models' in a classical textbook on artificial intelligence titled 'Artificial Intelligence: A Modern Approach (3rd edition).' Page 737 reads, 'A learning model that summarizes data with a set of parameters of fixed size (independent of the number of training examples) is called a parametric model.' In the revised version, we have corrected the statement as '…which summarize data with a fixed number of parameters regardless of sample size.'

Whether ANN is a parametric or nonparametric method is debatable. As the number of ANN parameters grows with the number of neurons/layers, ANN is considered as nonparametric in some references (e.g., Orr 1996, which is a basic and classical paper for radial basis function networks and has been cited more than 1000 times). This work follows Orr (1996) and considers RBF-ANN as a nonparametric model.

We agree with you that we also need to describe the advantages of these nonparametric models in dealing with nonlinear fitting, especially in our forest ecology nonlinear systems. Therefore, this sentence (L65) has been revised. The revised content in our manuscript now reads:

Traditional parametric methods, which summarize data with a fixed number of parameters regardless of sample size (e.g., logistic regression and perceptron) (Liu et al., 2020; Russell & Russell, 2016), usually require prior knowledge on a system to be modelled and assume a specific function for the system. Then the training data is used to estimate parameters of the presumed function. By comparison, nonparametric machine learning algorithms, which do not need sufficient understanding on the systems to be modelled, but seek the best fitting of the training data. They cannot be characterized by a fixed number of parameters (the number of parameters depends on the number of training examples, e.g., k-nearest neighbor, nonlinear support vector machine, and random forest). As a result, they are more elastic, can fit different function forms without prior knowledge, and achieve excellent fit performance (Russell and Norvig, 2016). Moreover, these nonlinear nonparametric models have advantages in dealing with nonlinear fitting and have great application potential in nonlinear systems such as forest ecology. Therefore, nonparametric machine learning algorithms may offer higher prediction accuracy in forest AGB estimation (Frey et al., 2019; Gleason and Im, 2012).

References

Frey, U. J., Klein, M., and Deissenroth, M.: Modelling complex investment decisions in Germany for renewables with different machine learning algorithms, Environmental Modelling & Software, 118, 61-75, https://doi.org/10.1016/j.envsoft.2019.03.006, 2019.

Gleason, C. J., and Im, J.: Forest biomass estimation from airborne LiDAR data using machine learning approaches, Remote Sensing of Environment, 125, 80-91, 2012.

Liu, B., Gao, L., Li, B., Marcos-Martinez, R., and Bryan, B. A.: Nonparametric machine-learning as an effective tool for mapping forest cover and exploring influential factors, Landscape Ecology, https://doi.org/10.1007/s10980-020-01046-0, 2020.

Orr, M. J. L.: Introduction to radial basis function networks. University of Edinburgh, 1996.

Russell, S. J, & Norvig, P.: Artificial intelligence: a modern approach, 3$^{rd}$ edition, Pearson Publisher, 2016.

**Comment 10:**  L68: remove 'nonparametric' and use non-linear

**Response to comment 10:** Thank you for reminding us of this point once again. We have revised it.

**Comment 11:**  L68-69: What do you mean with that the number of parameters on the number of training examples? This statement conflicts with your previous statement L65 and why or how does the number of parameters depend on n in kNN, SVM, and RF?

**Response to comment 11:** Thank you for this comment. The mistake we made in L65 has been revised as presented in the response to comment 9. The definitions of parametric and nonparametric models are sourced from Statistics and may be confused with some concepts in Computer Sciences. Here, we have carefully followed the definitions in classical textbook on artificial intelligence titled 'Artificial Intelligence: A Modern Approach (3rd edition)'. Page 737 reads, 'A nonparametric model is one that cannot be characterized by a bound set of parameters. For example, suppose that each hypothesis we generate simply retains within itself all of the training examples and uses all of them to predict the next example. Such a hypothesis family would be nonparametric because the effective number of parameters is unbounded — it grows with the number of examples. This approach is called instance-based learning or memory-based learning.' kNN, SVM, and RF all belong to this instance-based learning approach. Page 758 reads, 'Nonparametric models use all the data to make each prediction, rather than trying to summarize the data first with a few parameters. Examples include nearest neighbors and locally weighted regression.'

References

Russell, S. J, & Norvig, P.: Artificial intelligence: a modern approach, 3$^{rd}$ edition, Pearson
    Publisher, 2016.

**Comment 12:**  L70: restrict variable types? A linear regression is not restricted to specific data types. Actually, kNN and SVM require the same contrasts as a linear regression and only RF is able to handle non-contrasted categorical predictors

**Response to comment 12:** Thank you for correcting our mistake. We have revised it. The revised content in our manuscript now reads as shown in our **response to comment 9**.

**Comment 13:**  L71: What do you mean with the distribution of predictor variables? I think that kNN and SVM are indeed affected by the distribution of the predictors because they use distance measurements.

**Response to comment 13:** Thank you for your kind reminder. We have revised it. The revised content in our manuscript now reads as shown in our **response to comment 9**.

**Comment 14:**  L90: Why? Or at least provide a reference

**Response to comment 14:** Sorry, we made a mistake here. Thanks for bringing this to our attention. Our original intention was as follows: because of the spatial autocorrelation and spatial heterogeneity of forest AGB, its data distribution often does not conform to the assumption of the independent identical distribution of the traditional spatial statistical model. We have rewritten it as follows:

| Second, the data distribution of forest AGB is not consistent with the assumption of independent identical distribution required by the traditional spatial statistical model when it has spatial autocorrelation and spatial heterogeneity. |
|---|

**Comment 15:**  L97: Please provide example references

**Response to comment 15:** We agree with you that example references were necessary. We

have now cited some important references into our revised manuscript. The corresponding references are:

Qiu, Q. Y., Yun, G. L., Zuo, S. D., Yan, J., Hua, L. Z., Ren, Y., Tang, J. F., Li, Y. Y., Chen, Q.: Variations in the biomass of Eucalyptus plantations at a regional scale in southern China, Journal of Forestry Research, 29, 1263-1276, https://doi.org/10.1007/s11676-017-0534-0, 2018.

Kobler, J., Zehetgruber, B., Dirnböck, T., Jandl, R., Mirtl, M., & Schindlbacher, A.: Effects of aspect and altitude on carbon cycling processes in a temperate mountain forest catchment, Landscape Ecology, 34, 325-340, https://doi.org/10.1007/s10980-019-00769-z, 2019

**Comment 16:** L187: *'Each model was trained on 30 datasets. . .'* But within the CV, right? So it should be *'trained on 29 datasets'*.

**Response to comment 16:** Thank you for your correction. Since we have accepted one reviewer's suggestion on the cross-validation strategy, converted to adopt spatial block cross-validation strategy, we will rewrite Section 2.4.2 ('split datasets'). The original sentence of L187 will also be revised.

**Comment 17:** L199-200: Not exactly true, the activation function makes the transformation linear or non-linear. The fundamental matrix multiplication is a linear function.

**Response to comment 17:** Thank you for your kind reminder. We have revised it. The revised content in our manuscript now reads:

> The transformation from input space to hidden space is either linear or nonlinear, whereas the transformation from hidden space to output space is linear.

**Comment 18:** L197-199: Is there a reason you use an RFB-ANN? You could also use a normal DNN with relu activation functions and several hidden layers. Could you also explain the RBF-ANN in detail? I think that most users do not know how the RBF function is used by RBF-ANN.

**Response to comment 18:** We used RBF-ANN because it has been reported to have excellent performance in environmental applications, such as the spatial interpolation of soil moisture and nutrients (Xu, 2012), the prediction of river flow time series (Ghorbani et al., 2016), and the prediction of the energetic efficiency of a roughened solar air heater (Ghritlahre et al., 2018).

As suggested, we have further explained RBF-ANN in detail. Details of how the RBF function is used by RBF-ANN can be found at the following websites: (1) https://www.saedsayad.com/artificial_neural_network_rbf.htm, (2) https://pythonmachinelearning.pro/using-neural-networks-for-regression-radial-basis-function-networks/). We will add references to these websites in our manuscript and the Supplementary Material.

RBF-ANN is a three-layer neural network model which includes an input layer, a hidden layer, and an output layer. A Gaussian RBF is used inside the hidden-layer neurons of RBF-ANN. This differentiates an RBF-ANN from a regular neural network. First, each input vector is input into each basis. Then, the approximated function value is obtained by a simple weighted sum. Like any neural network, RNF-ANN is trained by back-propagation. Finally, the RBF network is implemented in a class and used to approximate a simple function. The transformation from input space to hidden space is either linear or nonlinear, whereas the transformation from hidden space to output space is linear. The function of the hidden layer is to map the vector from the indivisible low-dimensional linear state to the separable high-dimensional linear state so as to greatly accelerate the learning and convergence speed and avoid getting stuck in a local optimum (Elanayar and Shin, 1994; Xia and Xiu, 2007).

Elanayar, V. T. S., and Shin, Y. C.: Radial basis function neural network for approximation and estimation of nonlinear stochastic dynamic systems, IEEE Transactions on Neural Networks, 5, 594-603, 1994.

Ghorbani, M. A., Zadeh, H. A., Isazadeh, M., & Terzi, O.: A comparative study of artificial neural network (MLP, RBF) and support vector machine models for river flow prediction, Environmental earth sciences, 75(6), 476.1-476.14, https://doi.org/10.1007/s12665-015-5096-x, 2016.

Ghritlahre, H. K., & Prasad, R. K.: Exergetic performance prediction of solar air heater using MLP, GRNN and RBF models of artificial neural network technique, Journal of Environmental Management, 223(OCT.1), 566-575, https://doi.org/10.1016/j.jenvman.2018.06.033, 2018.

Xia, C. L., Xiu, J.: RBF ANN Nonlinear Prediction Model Based Adaptive PID Control of Switched Reluctance Motor, Proceedings of the CSEE, 27, 626-635, https://doi.org/10.1007/11893295_69, 2007.

Xu, Y.: Application of RBF artificial neural network with block effect to spatial interpolation of soil properties, Shuikexue Jinzhan/advances in Water Science, 23(1), 67-73, https://doi.org/10.14042/j.cnki.32.1309.2012.01.005, 2012.

**Comment 19:** L205: Wrong, RF can overfit, I thin with the law of large numbers you refer to the number of trees which is true that increasing the number of tree does not increase the generalization error. But if we assume that we have a sufficient number of trees, RF can overfit.

**Response to comment 19:** Thank you for your professional guidance in machine learning. The incorrect description has been deleted.

**Comment 20:** L227: What are these *"obvious advantages"*?

**Response to comment 20:** The detailed advantages of P-BSHADE compared with other traditional spatial statistics are described in the following paragraph. We have modified it to be more coherent and logical. The revised content in our manuscript now reads:

P-BSHADE is markedly different from the Kriging and Inverse Distance Weighting (IDW) algorithms. Compared with Kriging and IDW, the application of P-BSHADE to forest AGB interpolation has the following obvious advantages: (1) The spatial distribution of forest AGB is also characterized by spatial autocorrelation and heterogeneity, which have been taken into account in the P-BSHADE model. Taking into account spatial heterogeneity can effectively solve the difference in forest AGB distribution caused by different terrain or geographical location. However, Kriging and IDW only consider the spatial correlation between plots. (2) Additionally, P-BSHADE considers strongly correlated sample plots as neighboring plots, whereas the Kriging and IDW algorithms consider sites that are spatially close.

**Comment 21:** L248: 'when j=1,i=2,3,...30: when j=1,i=1,3,5,...,30)'What do you mean?

**Response to comment 21:** We made a mistake in L248. It should be 'when j=1, i=2,3,...30: when j=2, i=1,3,5,...,30'. The purpose of this passage was to illustrate the 'leave-one-out

cross-validation' situation. Given that we have changed the cross-validation strategy to spatial block cross-validation, we will revise this part.

**Comment 22:** L241-280: The description of the P-BSHADE model is too close to the original work Xu et al. 2013! Either you rewrite it completely in your own words, or which I suggest is that you try to summarize the method and move a detailed description to the Appendix. The P-BSHADE model is not the focus of your work.

**Response to comment 22:** Thank you for your kind advice. We have summarized the P-BSHADE method in the manuscript and have included detailed information in the Supplementary Material.

**Comment 23:** L312: 'A detailed description of the combined models. . .' is missing in the Supplementary Material

**Response to comment 23:** Thank you for your kind correction. Considering that the combination of the two models is the innovation of our manuscript, we included an introduction of how to combine the two methods in the main manuscript and deleted this sentence.

**Comment 24:** L347: Because of a low correlation you did not choose latitude, however, I hypothesize that the interaction between longitude and latitude has an effect on AGB, which you would not see in the correlation table unless to test explicitly the correlation between the interaction and the AGB.

**Response to comment 24:** Thank you for your kind advice. Using GeoDetector (http://www.geodetector.cn), we found that the interaction of longitude and latitude had an effect on AGB (see Table 1 below).

Table 1. Results of factor detector and interaction detector from GeoDetector.

|  | Longitude | Latitude | Interaction of longitude and latitude |
|---|---|---|---|
| q statistic | 0.54 | 0.09 | 0.75 |
| P value | 0.004 | 0.57 | Interaction result: enhanced, nonlinear |

We also tested the performance of models when longitude and latitude were both used as a predictor, and compared the results with those of models which only include longitude or which include neither longitude nor latitude (see Table 2 below). The results showed that using both longitude and latitude as predictors did not improve the performance of RF, SVM, or RBF-ANN. However, this does not mean that machine learning cannot capture the spatial relationships. We also show that the combination of three MLs and P-BSHADE can all improve the prediction accuracy compared to the single ML in all of the situations. We think that this further illustrates the advantages of the combined models.

Table 2. Performance of models in different predictor situations.

| Model | MAE | MRE | RMSE | nRMSE |
|---|---|---|---|---|
| Situation 1: includes longitude and latitude | | | | |
| SVM | 18.260 | 2.260 | 25.929 | 0.548 |
| RBF-ANN | 14.665 | 0.324 | 21.429 | 0.453 |
| RF | 16.435 | 1.292 | 27.419 | 0.579 |
| P-BSHADE | 12.069 | 0.251 | 19.490 | 0.412 |
| SVM & P-BSHADE | 5.075 | 0.108 | 6.495 | 0.137 |
| RBF-ANN & P-BSHADE | 12.164 | 0.244 | 17.546 | 0.371 |
| RF & P-BSHADE | 6.344 | 0.153 | 12.279 | 0.259 |
| Allometric Model | 16.412 | 0.577 | 25.080 | 0.530 |
| Situation 2: includes longitude but excludes latitude | | | | |
| SVM | 13.659 | 1.382 | 20.251 | 0.428 |
| RBF-ANN | 14.884 | 0.324 | 20.719 | 0.438 |
| RF | 14.919 | 1.107 | 21.371 | 0.451 |
| P-BSHADE | 12.069 | 0.251 | 19.490 | 0.412 |
| SVM & P-BSHADE | 6.879 | 0.128 | 10.757 | 0.227 |
| RBF-ANN & P-BSHADE | 12.597 | 0.260 | 17.819 | 0.376 |
| RF & P-BSHADE | 5.816 | 0.137 | 9.520 | 0.201 |
| Allometric Model | 16.412 | 0.577 | 25.080 | 0.530 |

| Situation 3: excludes longitude and latitude | | | | |
|---|---|---|---|---|
| SVM | 15.824 | 2.034 | 21.609 | 0.456 |
| RBF-ANN | 14.175 | 0.373 | 20.354 | 0.430 |
| RF | 15.201 | 1.092 | 21.330 | 0.451 |
| P-BSHADE | 12.069 | 0.251 | 19.490 | 0.412 |
| SVM & P-BSHADE | 9.679 | 0.173 | 16.055 | 0.339 |
| RBF-ANN & P-BSHADE | 12.227 | 0.232 | 18.124 | 0.383 |
| RF & P-BSHADE | 6.104 | 0.136 | 9.911 | 0.209 |
| Allometric Model | 16.412 | 0.577 | 25.080 | 0.530 |

Note: MAE: mean absolute error; MRE: mean relative error; RMSE: root mean square error; nRMSE: normalized root mean square error

**Comment 25:** L479: 'Machine learning models appear adept at tackling high-dimensional problems.' Yes, they do, but you do not have a 'high-dimensional problem'.

**Response to comment 25:** Thank you for your kind advice. We have deleted this inappropriate sentence.

**Comment 26:** L511: '. . . regression trees . . .' this applies only for RF and I wonder if the RF really suffer from a skewed response distribution. Do you have a reference?

**Response to comment 26:** Thank you for your kind suggestion. We have deleted this sentence.

**Comment 27:** L568: RF and SVM are also sensitive to hyper-parameters. It is myth that RF does not need hyper-parameter tuning (see https://doi.org/10.1002/widm.1301).

**Response to comment 27:** We are sorry for our arbitrary description of machine learning. It has been deleted. We have also read and cited the reference you kindly provided.

**Figures:**

**Comment 28:** Fig 3: The sematic figures of the ML do not fit to the way you used them in your work: RBF-ANN, RF, and SVM are illustrated as classifier with 4, 3, and 2 response classes, however, you use them as regression models (e.g. one output node for the RBF ANN, no majority voting for the RF etc.)

**Response to comment 28:** Thank you for your kind advice. We have modified Figure 3 as follows:

[Figure]

**Comment 29:** Fig 6: Revise your color choice, e.g. it is difficult to distinguish between the yellow and blue line.

**Response to comment 29:** Thank you for the kind advice. As suggested, we have revised Figure 6 in the revised manuscript.

[Figure]

We thank the reviewer and remain at your disposal for any further questions.

Yours sincerely,

Yin Ren

---

## Author Comment (AC5) · 5 Jun 2020

Dear Editor,

Thank you for your email and the reviewer's comments concerning our manuscript entitled "Improving maps of forest aboveground biomass: A combined approach using machine learning with a spatial statistical model" (ID: bg-2020-36). We thank the reviewer for the very helpful comments.

**Additional reply to reviewer's comments:**

Dear reviewer,

Anonymous Reviewer #2 recommended that we re-evaluate the performance of the machine learning models with additional predictors (at least latitude). For your convenience, the comments of anonymous reviewer #2 are listed below:

*"b) They claim that the joint model combines the advantages of ML and the P-BSHADE model, the predictive non-linearity advantage of ML and the ability of the P-BSHADE to capture spatial relationships. However, if they are given the chance I think that ML models are also capable of detecting and using spatial relationships, that means, you have to provide them not only longitude but also latitude as predictor! Based on correlation with AGB, the authors selected only longitude, however, I would assume that an interaction of longitude and latitude would be a good predictor of spatial relationships (two variables of an interaction can show by themselves low correlation). Moreover, ML models such as RF are outstanding in detecting interactions and higher-order interactions (if they are given the chance). Also, hyper-parameter tuning is important in ML to improve predictive performance, even for RF! (e.g. see Probest et al., 2019 https://doi.org/10.1002/widm.1301). I recommend that the authors re-evaluate the performance of the ML models with hyper-parameter tuning, nested cross-validation, and additional predictors (at least latitude)."*

We think this comment is relate to your General Comment 2). For your convenience, your comment is presented here:

*"2) The cross-validation strategy that you used is not suitable if you have spatially clustered data (as you obviously have looking at the map). This is shown by several studies (see references below, to mention just a few). What would be appropriate is a spatial cross-validation that is testing the ability of your model to make predictions for spatially new samples. At least you should take care that you never use data points from the same forest patch for both training and testing. Otherwise, it is not possible to evaluate the ability of your models for regional mapping. Including coordinates as predictors when the data are spatially clustered is very dangerous (see Meyer et al 2019) and can lead to high overfitting which can only be revealed with spatial cross-validation. So I recommend that in addition to spatial cross-validation, to perform a spatial variable selection (i.e. can the predictors be used to make predictions for new locations?)*

*1. Meyer, H., Reudenbach, C., Wöllauer, S., Nauss, T., 2019. Importance of spatial predictor variable selection in machine learning applications - Moving from data reproduction to spatial prediction. Ecological Modelling. 411, 108815.*

2. Pohjankukka, J., Pahikkala, T., Nevalainen, P., Heikkonen, J., 2017. Estimating the prediction performance of spatial models via spatial k-fold cross validation. Int. J. Geogr. Inform. Sci. 31, 2001–2019. https://doi.org/10.1080/13658816.2017. 1346255.

3. Roberts, D.R., Bahn, V., Ciuti, S., Boyce, M.S., Elith, J., Guillera-Arroita, G., Hauenstein, S., Lahoz-Monfort, J.J., Schröder, B., Thuiller, W., Warton, D.I., Wintle, B.A., Hartig, F., Dormann, C.F., 2017. Cross-validation strategies for data with temporal, spatial, hierarchical, or phylogenetic structure. Ecography.

4. Valavi, R., Elith, J., Lahoz-Monfort, J.J., Guillera-Arroita, G., 2018. blockcv: an r package for generating spatially or environmentally separated folds for k-fold cross validation of species distribution models. BioRxiv.*

To test whether include or exclude latitude, and whether leave-one out cross-validation or spatial block cross-validation would have impact on models performance in the forest AGB estimation, we re-build the model and evaluate the results again. We found that there has a error on Figure 1 to 3 in the previous response to your comments (https://www.biogeosciences-discuss.net/bg-2020-36/bg-2020-36-AC3-supplement.pdf). All the results were correct in this response. Therefore, we revised and presented the final results including six situations in Table 1:

The results from comparing situation 2 and 5 showed that cross validation strategy has significant impact on the ML models and P-BSHADE model, but showed limited impact on the three combination models. We agree with your comments and suggestions and decided to adopt spatial block cross-validation.

The results showed that using both longitude and latitude as predictors did not improve the performance of RF, SVM, or RBF-ANN. However, this does not mean that machine learning cannot capture the spatial relationships. We also show that the combination of three MLs and P-BSHADE can all improve the prediction accuracy compared to the single ML in all of the situations. We think that this further illustrates the advantages of the combined models.

Table 1 Performance of models in different predictor situations

| Model | MAE | MRE | RMSE | nRMSE |
|---|---|---|---|---|
| Situation 1: includes longitude and latitude with spatial block cross-validation | | | | |
| SVM | 18.260 | 2.260 | 25.929 | 0.548 |
| RBF-ANN | 14.665 | 0.324 | 21.429 | 0.453 |
| RF | 16.435 | 1.292 | 27.419 | 0.579 |
| P-BSHADE | 12.069 | 0.251 | 19.490 | 0.412 |
| SVM & P-BSHADE | 5.075 | 0.108 | 6.495 | 0.137 |
| RBF-ANN & P-BSHADE | 12.164 | 0.244 | 17.546 | 0.371 |
| RF & P-BSHADE | 6.344 | 0.153 | 12.279 | 0.259 |
| Allometric Model | 16.412 | 0.577 | 25.080 | 0.530 |
| Situation 2: includes longitude but excludes latitude with spatial block cross-validation | | | | |
| SVM | 13.659 | 1.382 | 20.251 | 0.428 |
| RBF-ANN | 14.884 | 0.324 | 20.719 | 0.438 |
| RF | 14.919 | 1.107 | 21.371 | 0.451 |
| P-BSHADE | 12.069 | 0.251 | 19.490 | 0.412 |
| SVM & P-BSHADE | 6.879 | 0.128 | 10.757 | 0.227 |
| RBF-ANN & P-BSHADE | 12.597 | 0.260 | 17.819 | 0.376 |
| RF & P-BSHADE | 5.816 | 0.137 | 9.520 | 0.201 |
| Allometric Model | 16.412 | 0.577 | 25.080 | 0.530 |
| Situation 3: excludes longitude and latitude with spatial block cross-validation | | | | |
| SVM | 15.824 | 2.034 | 21.609 | 0.456 |
| RBF-ANN | 14.175 | 0.373 | 20.354 | 0.430 |
| RF | 15.201 | 1.092 | 21.330 | 0.451 |
| P-BSHADE | 12.069 | 0.251 | 19.490 | 0.412 |
| SVM & P-BSHADE | 9.679 | 0.173 | 16.055 | 0.339 |
| RBF-ANN & P-BSHADE | 12.227 | 0.232 | 18.124 | 0.383 |
| RF & P-BSHADE | 6.104 | 0.136 | 9.911 | 0.209 |
| Allometric Model | 16.412 | 0.577 | 25.080 | 0.530 |
| Situation 4: includes longitude and latitude with leave-one out cross-validation | | | | |
| SVM | 8.208 | 0.220 | 11.064 | 0.232 |
| RBF-ANN | 12.710 | 0.268 | 19.466 | 0.409 |
| RF | 10.696 | 0.289 | 20.012 | 0.420 |
| P-BSHADE | 12.045 | 0.279 | 19.781 | 0.415 |
| SVM & P-BSHADE | 5.214 | 0.110 | 6.668 | 0.140 |
| RBF-ANN & P-BSHADE | 11.876 | 0.224 | 17.479 | 0.367 |
| RF & P-BSHADE | 5.853 | 0.145 | 10.963 | 0.230 |

| Situation 5: includes longitude and excludes latitude with leave-one out cross-validation(in submitted article) | | | | |
|---|---|---|---|---|
| SVM | 11.168 | 0.248 | 10.388 | 0.218 |
| RBF-ANN | 12.149 | 0.267 | 10.388 | 0.218 |
| RF | 10.155 | 0.259 | 9.428 | 0.198 |
| P-BSHADE | 18.371 | 0.391 | 14.077 | 0.296 |
| SVM & P-BSHADE | 6.883 | 0.125 | 6.304 | 0.132 |
| RBF-ANN & P-BSHADE | 10.136 | 0.205 | 9.633 | 0.202 |
| RF & P-BSHADE | 5.679 | 0.130 | 5.299 | 0.111 |
| Situation 6: excludes longitude and latitude with leave-one out cross-validation | | | | |
| SVM | 11.473 | 0.240 | 18.426 | 0.387 |
| RBF-ANN | 12.674 | 0.269 | 18.584 | 0.390 |
| RF | 10.759 | 0.232 | 17.043 | 0.358 |
| P-BSHADE | 12.045 | 0.279 | 19.781 | 0.415 |
| SVM & P-BSHADE | 9.203 | 0.157 | 15.977 | 0.336 |
| RBF-ANN & P-BSHADE | 16.282 | 0.395 | 27.770 | 0.583 |
| RF & P-BSHADE | 5.933 | 0.122 | 9.765 | 0.205 |

Note: MAE: mean absolute error; MRE: mean relative error; RMSE: root mean square error; nRMSE: normalized root mean square error

We thank the reviewer and remain at your disposal for any further questions.

Yours sincerely,
Yin Ren